# Margins are Insufficient for Explaining Gradient Boosting

**Allan Grønlund** *†            **Lior Kamma** †            **Kasper Green Larsen** †

## Abstract

Boosting is one of the most successful ideas in machine learning, achieving great practical performance with little fine-tuning. The success of boosted classifiers is most often attributed to improvements in margins. The focus on margin explanations was pioneered in the seminal work by Schapire et al. (1998) and has culminated in the $k$'th margin generalization bound by Gao and Zhou (2013), which was recently proved to be near-tight for some data distributions (Grønlund et al. 2019). In this work, we first demonstrate that the $k$'th margin bound is inadequate in explaining the performance of state-of-the-art gradient boosters. We then explain the short comings of the $k$'th margin bound and prove a stronger and more refined margin-based generalization bound for boosted classifiers that indeed succeeds in explaining the performance of modern gradient boosters. Finally, we improve upon the recent generalization lower bound by Grønlund et al. (2019).

## 1 Introduction

Boosting is a powerful technique for producing highly accurate voting classifiers by combining less accurate base learners. Boosting algorithms are typically easy to fine tune and obtain state-of-the-art performance on many learning tasks. Boosting dates back to the seminal work introducing the AdaBoost algorithm [4] and much work has gone into understanding and developing better boosting algorithms. The best performing boosting algorithms are typically variants of gradient boosters [5], such as LightGBM [10] and XGBoost [2], using Regression Trees as base learners.

Classic experiments [14] showed that boosting algorithms tend to improve their test accuracy even when training past the point of perfectly classifying the training data. This may seem counter-intuitive, as adding more base learners, results in a more complex model, that hence might be more prone to overfitting. This phenomenon is often explained by observed improvements in margins. For binary classification with a sample space $\mathcal{X}$, labels in $\{-1, 1\}$ and a class of base learners $\mathcal{H} \subseteq \mathcal{X} \to [-1, 1]$, a voting classifier $f : \mathcal{X} \to \{-1, 1\}$ has the form $f(x) = \text{sign}(\sum_{h \in \mathcal{H}} \alpha_h h(x))$ with all $\alpha_h \geq 0$. A voting classifier thus takes a weighted "vote" among the base learners to obtain its prediction. When speaking of margins, we assume $\sum_h \alpha_h = 1$, which can always be achieved by rescaling the $\alpha$'s by their sum without changing $f$. The margin of a training point $(x, y)$ with $x \in \mathcal{X}$ and $y \in \{-1, 1\}$ is then defined as $y \sum_h \alpha_h h(x)$. The margin is thus a value in $[-1, 1]$ which is positive when $f(x) = y$ and negative otherwise. Intuitively, large (positive) margins mean that $f$ is not only correct but very certain in its predictions. Margin theory, starting with the work of Schapire et al. [4], formalized this by proving generalization bounds demonstrating that large margins imply better generalization. It was also shown that the theoretical generalization bounds fit very well with the observed behavior of AdaBoost that tends to keep improving margins even when training past the point of perfectly classifying the training data [14].

However, shortly after [4] and [14] was published, Breiman [1] proved a generalization bound based on the minimal margin (the smallest margin achieved by a training point) that was sharper than the generalization bound in Schapire et al. [4]. He then designed a new boosting algorithm, named *Arc-GV*, that provably optimizes the minimal margin, which AdaBoost does not (see [7] for the full story of maximizing the minimal margin). In the same paper, Breiman experimentally showed that Arc-GV produced not just a better minimal margin, but better margins overall, than AdaBoost. However, AdaBoost still obtained a better generalization and test error. This seemed to contradict margin theory, as according to margin theory, all other things being equal, then larger margins should imply better generalization. Later it was shown by Reyzin and Schapire [13] that Breiman's experiments did not accurately take into account the complexity of the base learner trees created by AdaBoost and Arc-GV, as repeating the experiments showed that Arc-GV produced trees of larger depth than AdaBoost, and deeper trees may be more prone to overfitting. Reyzin and Schapire then considered the same experiments using stumps as base learners, forcing identical depth trees between the algorithms, and in this case, AdaBoost produced better margin distributions than Arc-GV and also generalized better. These findings support the view that better margins provide better generalization as presented in [4, 14].

Later, [16, 11, 6] showed improved generalization bounds that subsumed both the generalization bounds by Schapire et al., and Breiman, providing further theoretical support for margin theory. The current strongest generalization bounds are as follows. Let $\mathcal{D}$ be any distribution over $\mathcal{X} \times \{-1, 1\}$ and define $\mathcal{L}_{\mathcal{D}}(f) = \Pr_{(x,y) \sim \mathcal{D}}[f(x) \neq y]$ as the out-of-sample error of a voting classifier $f$. Also, for a set $S = \{(x_i, y_i)\}_{i=1}^{m}$ of $m$ labeled samples drawn i.i.d. from $\mathcal{D}$, define $\mathcal{L}_{S}^{\theta}(f) = \Pr_{(x,y) \sim S}[yf(x) < \theta]$ as the fraction of points in $S$ with margin less than $\theta$ (the notation $(x, y) \sim S$ denotes a uniform random point $(x, y)$ in $S$). With this notation, there are two strongest current generalization bounds. The first [11] uses Rademacher complexity to show that with high probability over the sample set $S$, it holds for every margin $\theta \in (0, 1]$ and every voting classifier $f$ that:

$$\mathcal{L}_{\mathcal{D}}(f) \leq \mathcal{L}_{S}^{\theta}(f) + O\left(\sqrt{\frac{\lg |\mathcal{H}|}{\theta^2 m}}\right). \tag{1}$$

The $k$'th margin bound by Gao and Zhou [6] improves this for $\mathcal{L}_{S}^{\theta}(f) = o(1/\lg m)$ and is as follows:

$$\mathcal{L}_{\mathcal{D}}(f) \leq \mathcal{L}_{S}^{\theta}(f) + O\left(\frac{\lg |\mathcal{H}| \lg m}{\theta^2 m} + \sqrt{\mathcal{L}_{S}^{\theta}(f) \cdot \frac{\lg |\mathcal{H}| \lg m}{\theta^2 m}}\right). \tag{2}$$

The $k$'th margin bound subsumes both Breiman's min margin generalization bound and the original generalization bound by Schapire et al. For infinite $\mathcal{H}$, one may replace $\lg |\mathcal{H}|$ in the above bounds with the VC-dimension of $\mathcal{H}$ times a $\lg m$ factor (as is standard). For simplicity, we focus on the case of finite $\mathcal{H}$ throughout the paper. Moreover, recent work by Grønlund et al. [8] shows that the margin bounds above are near-tight. Formally, they show that for (almost) all margins $\theta$, there exists a data distribution $\mathcal{D}$ and a set of base learners $\mathcal{H}$, such that with constant probability over the sample set $S$, there is a voting classifier $f$ such that

$$\mathcal{L}_{\mathcal{D}}(f) \geq \mathcal{L}_{S}^{\theta}(f) + \Omega\left(\frac{\lg |\mathcal{H}| \lg m}{\theta^2 m} + \sqrt{\mathcal{L}_{S}^{\theta}(f) \cdot \frac{\lg |\mathcal{H}|}{\theta^2 m}}\right). \tag{3}$$

Moreover, the lower bound holds for any value of $\mathcal{L}_{S}^{\theta}(f) \leq 49/100$ and any value of $\lg |\mathcal{H}|$ [8].

**Remark.** Many boosting algorithms produce classifiers $f = \sum_h \alpha_h h$ where $\sum_h \alpha_h \neq 1$ or where base learners output values in $\mathbb{R}$ rather than $[-1, 1]$. To apply margin theory, following [15], such classifiers are rescaled as follows: For each $h$ with output range $[a_h, b_h]$ and coefficient $\alpha_h$, divide all outputs of $h$ by $\Delta_h = \max\{|a_h|, |b_h|\}$ and multiply $\alpha_h$ by $\Delta_h$. Afterwards, divide all $\alpha_h$ by $\sum_h \alpha_h$.

## 1.1  Our contribution.

**A new margin lower bound:**   Comparing the current best upper and lower bounds, we see that (2) and (3) match when $\mathcal{L}_{S}^{\theta}(f)$ approaches 0. Similarly, we see that (2) and (1) match as $\mathcal{L}_{S}^{\theta}(f)$ approaches a constant. But what is the true behavior in-between? The $k$'th margin bound (2) gained the factor $\mathcal{L}_{S}^{\theta}(f)$ inside the $\sqrt{\cdot}$ but lost a factor $\lg m$ compared to (1). Can the $\lg m$ factor be removed?

What is the correct behavior as $\mathcal{L}_S^\theta(f)$ goes from 0 towards 1? In this work, we show an improved generalization lower bound of:

$$\mathcal{L}_\mathcal{D}(f) \geq \mathcal{L}_S^\theta(f) + \Omega\left(\frac{\lg|\mathcal{H}|\lg m}{\theta^2 m} + \sqrt{\mathcal{L}_S^\theta(f) \cdot \frac{\lg|\mathcal{H}|\lg(\mathcal{L}_S^\theta(f)^{-1})}{\theta^2 m}}\right). \tag{4}$$

Our lower bound shows that the $\lg m$ factor inside the $\sqrt{\cdot}$ has to show up as $\mathcal{L}_S^\theta(f)$ drops to $m^{-\varepsilon}$ for any constant $\varepsilon > 0$. Moreover, our new lower bound completely settles the generalization performance of boosting in terms of margins whenever $\mathcal{L}_S^\theta(f)$ is outside the range $m^{-o(1)}$ to $o(1)$. It also nicely interpolates between the $\mathcal{L}_s^\theta(f) = 0$ and $\mathcal{L}_S^\theta(f) = 1$ case. We conjecture that the lower bound gives the correct margin-based tradeoff, i.e. that it is possible to improve the upper bounds (1) and (2) to match (4). Our proof is based on the work in [8], and the recent near-tight generalization lower bound proof for Support Vector Machines shown in [9]. Due to lack of space, the full formulation of our lower bound and the proof can be found in the full version of the paper.

**A new refined margin generalization bound:** The main part of our paper considers a new refined margin based generalization bound for voting classifiers (boosting algorithms). First, we present experiments showing that the classic margin bounds alone fail to explain the performance of state-of-the art gradient boosting algorithms. More concretely, we show that gradient boosters actually may produce smaller and smaller margins when run for many iterations, despite the test accuracy staying the same or even improving. We additionally demonstrate that the classic version of AdaBoost may produce significantly better margins than gradient boosters, despite gradient boosters obtaining similar or even better test accuracy and generalization error than AdaBoost. To explain this inconsistency, we observe experimentally that the trees produced by gradient boosters return very small values on all but a few training points, thus making minimal changes to most predictions when added to the voting classifier. We then use this insight to prove a new margin-based generalization bound for boosting algorithms which also take into account the magnitude of predictions by base learners. Finally, we run experiments demonstrating that our refined generalization bounds in fact succeed in explaining and predicting the performance of boosting algorithms. In addition to achieving a better theoretical understanding of boosting algorithms, in particular gradient boosters, these new insights may potentially lead to new algorithms with better accuracy by using regularization inspired by our new generalization bound or more directly optimizing it.

## 2   Insufficiency of current margin bounds

From the margin-based upper and lower bounds, it may seem that we have all the theory necessary for understanding the generalization performance of boosters. To confirm the theory, we ran experiments with AdaBoost and the state-of-the-art gradient booster LightGBM on standard data sets with the same size trees as base learners. For all experiments we only change the tree size and learning rate of the LightGBM hyperparameters. For AdaBoost we allow the same tree size, unlimited depth, as well as forcing a minimum number of elements in each tree learner to be 20 as is default in LightGBM.

Figure 1b shows a plot of the margin distributions for the two boosters trained on the Forest Cover dataset. From this plot, it is obvious that AdaBoost achieves significantly better margins than LightGBM. Indeed, the $k$'th smallest margin of AdaBoost, is much larger than the $k$'th smallest margin of LightGBM for all $k$ where at least one of the two margins are non-negative. Thus, from the generalization bounds (1) and (2), AdaBoost should have a much smaller out-of-sample error than LightGBM. However, the corresponding test errors in Figure 1a show a very different story, with LightGBM slightly outperforming AdaBoost. Furthermore, as shown in Section 3, the trees produced by LightGBM are in fact deeper than the trees produced by AdaBoost. This gives rise to some concerns regarding the explanatory power of margins. To further underline the theoretical inconsistency, we examine the two generalization bounds (1) and (2). When applying the generalization bounds to AdaBoost and LightGBM, then for any choice of $p = \mathcal{L}_S^\theta(f) \in [0,1]$, the only parameter that vary between AdaBoost and LightGBM is $\theta^{-2}$. That is, if we choose $\theta$ as the $(pm)$'th smallest margin, i.e. fix $\mathcal{L}_S^\theta(f) = p$, then only the value of $\theta$ differ between the two boosters and the generalization error grows as $\theta^{-2}$. Figure 2a shows a plot of $\theta^{-2}$ as a function of $\mathcal{L}_S^\theta(f)$ for the two boosters. Clearly the penalty in the generalization error is much smaller for AdaBoost, suggesting that AdaBoost should perform much better than LightGBM, despite the test errors in Figure 1a showing that LightGBM outperforms AdaBoost. To investigate this phenomenon

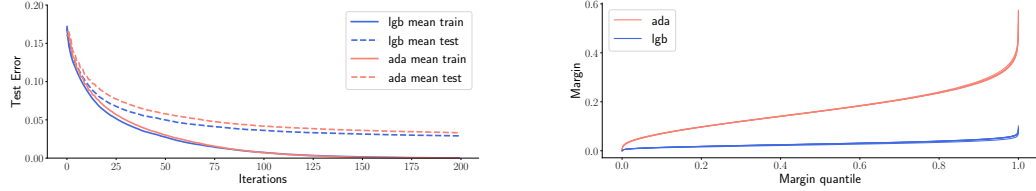

(a) Mean training and test error over five runs. The standard deviation of the final test error is 0.00037 for AdaBoost and smaller for LightGBM.

(b) Sorted margin values.

Figure 1: Accuracy and margin plots for AdaBoost and LightGBM on the Forest Cover data set.

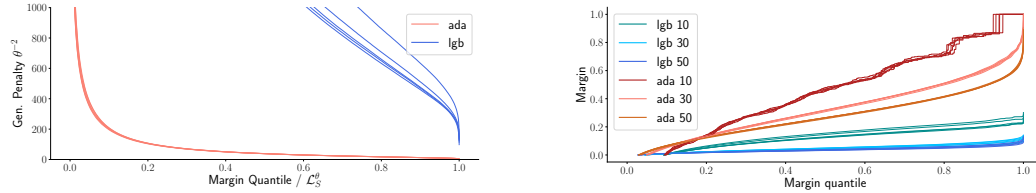

(a) Plot of $\theta^{-2}$ when choosing $\theta$ as the $(pm)$'th smallest margin for $p \in [0,1]$. The margins are those also shown in Figure 1b.

(b) Development in margin distributions for AdaBoost and LightGBM.

Figure 2: Generalization penalties and margin distributions on the Forest Cover data set.

further, we have plotted the margin distribution of the two boosters after $t = 10, 20$ and $50$ iterations of training, see Figure 2b. It is clear from this plot that the margins of the gradient booster, learned by LightGBM, deteriorate quickly with the number of training iterations. To explain why the margins quickly drops towards 0 for the gradient booster, we take a closer look at the trees produced by LightGBM compared to AdaBoost. Figure 3 shows a histogram of the predictions made by the trees produced by LightGBM. It is very striking from this histogram that the trees making up the LightGBM gradient booster makes very small (in absolute value) predictions on most data points, whereas AdaBoost always makes predictions in $\{-1, 1\}$. Note that each tree always has its largest prediction among $\{-1, 1\}$. Thus, LightGBM produces trees that only significantly change the predictions of very few data points, while leaving almost all others unchanged. When training more and more trees, this causes the margins to diminish. To see this, consider as an example a training point $(x, 1)$ and assume the first trained tree $h$ makes a (correct) prediction of $h(x) = 1$ and is assigned a weight of $\alpha_h = 1$. After the first training iteration, the margin of $(x, 1)$ is 1. However, as training progresses, many more trees may be produced that all predict 0 on $x$ while being assigned a weight of 1. Since margins are normalized, $\sum_{h \in \mathcal{H}} \alpha_h = 1$, this means that the margin of $x$ drops to $1/t$ after $t$ rounds of training. The drop in predicted accuracy by the generalization bounds (1) and (2) seem unreasonable if we think about the data point $x$ (the error is expected to grow as $t^2$ or $t$). A

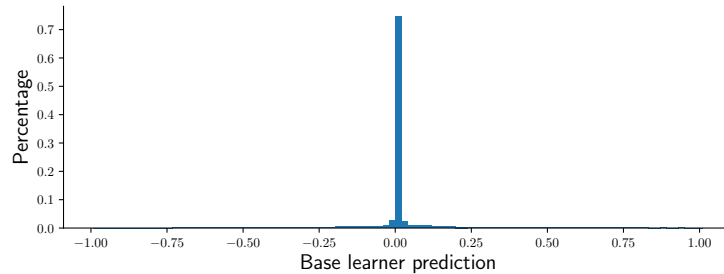

Figure 3: Histogram of base learner predictions for LightGBM on the Forest Cover data set. Only about 1 in 5000 predictions are larger than 0.95 in absolute value.

possible explanation of the shortcomings of current generalization bounds is thus that they simply

treat base learners as arbitrary functions in $\mathcal{X} \to [-1, 1]$. That is, they pay no attention to the fact that base learners trained by gradient boosters make very small predictions on almost all data points. To further support this claim, we note that the proof of the previous generalization lower bound (3) as well as our improved bound (4) construct a set of base learners $\mathcal{H}$ where all $h \in \mathcal{H}$ make predictions among $\{-1, 1\}$, i.e. they make no predictions of small magnitude. This further supports the belief that an explanation based on the magnitude of predictions may be found, which is the focus of the next section. We have used a tree size of 256 as large tree sizes are used in practice and provide better test errors. Furthermore, the phenomena we are studying is clearer for large tree sizes. In Section 3 we show results for both large trees and stumps. We note that base learners with real valued predictions were first considered by Schapire and Singer [15] that generalized the generalization bound of Schapire et al. [14] to work with real values but without otherwise changing the bound.

## 3  Refined margin bounds

Motivated by the empirical observations in the previous section, we prove a more refined margin based generalization bound for voting classifiers. Define from a voting classifier $f$ the notation $\Delta(x, h) := |f(x) - h(x)|$. Intuitively, if a voting classifier $f$ has a small margin on a training point $x$, but this is the result of using mostly base learners $h$ that make small predictions (in absolute value), then $\Delta(x, h)$ will be small for most $h$ in $f$. Also define from a voting classifier $f = \sum_h \alpha_h h$ the distribution $\mathcal{Q}(f)$ over base learners, which simply returns $h$ with probability $\alpha_h$. With this notation, our new generalization bound states that for any distribution $\mathcal{D}$ over $\mathcal{X} \times \{-1, 1\}$ and for any margin $\theta$, it holds with high probability over a set $S \sim \mathcal{D}^m$ that all voting classifiers $f$ satisfy:

$$\mathcal{L}_{\mathcal{D}}(f) \leq \mathcal{L}_S^{\theta}(f) + O\left(\frac{N \lg |\mathcal{H}| \lg m}{m} + \sqrt{\mathcal{L}_S^{\theta}(f) \cdot \frac{N \lg |\mathcal{H}| \lg m}{m}}\right), \tag{5}$$

where $N = \max\{\theta^{-2} \cdot \left(\mathbb{E}_{(x,y)\sim S}\left[\mathbb{E}_{h \sim \mathcal{Q}(f)}\left[\Delta(x, h)^2\right]^{(\lg(16m))/2}\right]\right)^{2/(\lg(16m))}, \theta^{-1}\}$.

**Never worse.**  Comparing our bound to the $k$'th margin bound (2), we see that (5) equals the $k$'th margin bound when $N = \Theta(\theta^{-2})$. First, we argue that we always have $N = O(\theta^{-2})$, i.e. (5) is never worse than the $k$'th margin bound. To see this, observe that $\Delta(x, h) \leq 2$ since all $h \in \mathcal{H}$ produce values in $[-1, 1]$. Thus, $\Delta(x, h)^2 \leq 4$ and $\mathbb{E}_{h \sim \mathcal{Q}(f)}\left[\Delta(x, h)^2\right] \leq 4$. This implies $\left(\mathbb{E}_{(x,y)\sim S}\left[\mathbb{E}_{h \sim \mathcal{Q}(f)}\left[\Delta(x, h)^2\right]^{(\lg(16m))/2}\right]\right)^{2/\lg(16m)} \leq 4$, hence we always have $N = O(\theta^{-2})$.

**Potentially much better.**  Next, we demonstrate that our new bound may be significantly better than previous generalization bounds for very natural voting classifiers. For any desired margin $\theta \in (0, 1]$, consider an example of a voting classifier $f(x) = \sum_{i=1}^{1/\theta} \theta h_i(x)$ such that for each training point $(x, y)$, there is exactly one hypothesis $h_i$ with $h_i(x) = y$ and all others have $h_j(x) = 0$. This example is quite similar to the empirical performance of LightGBM seen in Section 2, where most hypotheses make small predictions on most training points. The voting classifier $f$ has a margin of $\theta$ on all training points and thus the $k$'th margin bound predicts a generalization error of $O(\lg |\mathcal{H}| \lg m/(m\theta^2))$ (since $\mathcal{L}_S^{\theta}(f) = 0$ when all points have margin $\theta$). Let us now estimate $N$ in (5). First, fix an $(x, y) \in S$ and consider the expression $\mathbb{E}_{h \sim \mathcal{Q}(f)}\left[\Delta(x, h)^2\right]^{(\lg(16m))/2} = \left(\sum_{i=1}^{1/\theta} \theta \cdot \Delta(x, h_i)^2\right)^{(\lg(16m))/2} = \left(\theta \cdot (1 - \theta)^2 + (1 - \theta)\theta^2\right)^{(\lg(16m))/2} < \theta^{(\lg(16m))/2}$. Since this holds for every $(x, y)$, we have $\left(\mathbb{E}_{(x,y)\sim S}\left[\mathbb{E}_{h \sim \mathcal{Q}(f)}\left[\Delta(x, h)^2\right]^{(\lg(16m))/2}\right]\right)^{2/\lg(16m)} < \theta$. Plugging that into the definition of $N$, we see that $N \leq \max\{\theta^{-2} \cdot \theta, \theta^{-1}\} = \theta^{-1}$. That is, the dependency on the margin has improved by a factor $\theta$ and our new generalization bound predicts $\mathcal{L}_{\mathcal{D}}(f) = O(\lg |\mathcal{H}| \lg m/(m\theta))$.

**Comparison to earlier work.**  In recent work, Cortes et al. [3], also proved refined generalization bounds for gradient boosters. Their works shows, that if the $q$-norm of the vector of leaf predictions for each tree trained by a gradient booster is small, then the trees have smaller VC-dimension and hence the voting classifier has better generalization performance (by using previous generalization bounds). Note that their bound only depends on the leaf predictions and does not take into account

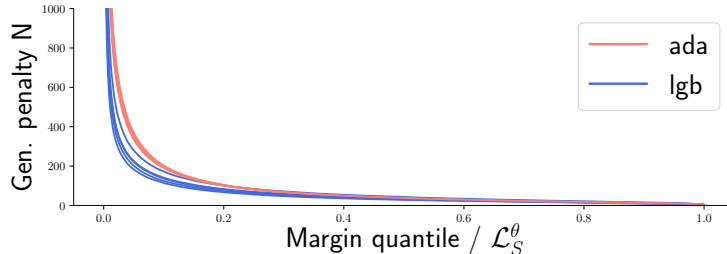

Figure 4: Generalization penalty $N$ on the Forest Cover data set when choosing $\theta$ as the $(pm)$'th smallest margin for $p \in [0, 1]$.

the number of training points in each leaf. Our experiment in Figure 3 shows that for each base learner, only a tiny fraction (about 1 in 5000) of training points end in a leaf with large prediction, which our bound takes into account.

Table 1: Comparing AdaBoost with LightGBM. In this experiment the trees used as bare learners are of increasing size relative to the data size. Each value shown is the average over several runs and each run use 200 rounds of boosting. Moment is $\left( \mathbb{E}_{(x,y) \sim S} \left[ \mathbb{E}_{h \sim \mathcal{Q}(f)} \left[ \Delta(x, h)^2 \right]^{(\lg(m))/2} \right] \right)^{2/\lg(m)}$.

| Data Set | Alg. | Train Err | Test Err | Mean Margin | Max Depth | Mean Depth | Moment |
|----------|------|-----------|----------|-------------|-----------|------------|--------|
| Forest | ada | 0.0001 | 0.0331 | 0.1696 | 22.0 | 12.4 | 0.969 |
| | lgb | 0.0002 | 0.0291 | 0.0280 | 23.7 | 13.9 | 0.025 |
| Boone | ada | 0.00009 | 0.0589 | 0.311 | 17.5 | 10.2 | 0.917 |
| | lgb | 0.00009 | 0.0552 | 0.0818 | 17.6 | 10.4 | 0.0564 |
| Higgs | ada | 0.178 | 0.277 | 0.0747 | 24.9 | 13.5 | 0.99 |
| | lgb | 0.185 | 0.251 | 0.018 | 26 | 14.7 | 0.0289 |
| Diabetes | ada | 0 | 0.268 | 0.148 | 3.5 | 2.63 | 0.973 |
| | lgb | 0.0264 | 0.26 | 0.142 | 3.5 | 2.63 | 0.214 |

**Empirical evaluation.** Our new generalization bound carefully takes the magnitude of predictions made by the base learners into account, thus there is hope that (5) may better explain the experiments in the previous section. To test this, we have run the experiments again, this time plotting the value of $N$ as a function of $p = \mathcal{L}_S^\theta(f)$. That is, we notice that for two voting classifiers produced by AdaBoost and LightGBM, respectively, the only thing that varies in (5) when choosing the $(pm)$'th smallest margin, i.e. $p = \mathcal{L}_S^\theta(f)$, is the value of $N$. Thus smaller values of $N$ imply better generalization according to the theory. Figure 4 shows the result of the experiment. Quite remarkably, the relative ordering of AdaBoost and LightGBM match the observed test errors from Figure 1a much better, i.e. LightGBM slightly outperforms AdaBoost. We have repeated the same experiment on more data sets and summarized the results in Table 1. The parameters for the experiments are shown in Table 2

Table 2: Data sets, all freely available, and parameters considered in the experiments. LR means learning rate as used in LightGBM. For each experiment we randomly split the data set in half to get a training set and a test set of equal size. For the Higgs dataset of size 11 million, we sample a subset of 2 million data points that we randomly split evenly into train and test set. For Forest Cover only the first two classes are used to make it into a binary classification problem.

| Data Set | Data Size | Tree Size | LR | Stumps LR | Runs |
|----------|-----------|-----------|-----|-----------|------|
| Diabetes | 768 | 5 | 0.1 | 0.1 | 100 |
| Boone | 65032 | 96 | 0.2 | 0.6 | 5 |
| Forest Cover | 495141 | 256 | 0.3 | 0.3 | 5 |
| Higgs | 2000000 | 512 | 0.3 | 0.3 | 5 |

In all experiments, the margin distribution, here represented by the mean margin, is much worse for the LightGBM classifier, while the height of the trees used, both the max height and the mean height, is larger. Still the LightGBM classifier generalizes at least as well (in fact, slightly better) than the AdaBoost classifier. Table 1 also shows that the moment value from our generalization bound is significantly better for the LightGBM classifier. When we consider our new generalization bound, the theory nicely matches the observed test errors in the same way as was shown in Figure 4 for all data sets. While not final proof that this is the real or only explanation, it suggests that the success of gradient boosters, despite having poor margins, may be explained by the many small predictions made by the base learner trees. The standard deviations of the test statistics are left out since they are extremely small for the three large data sets (and we have run 100 iterations of the small Diabetes data set). For completeness we have included the same experiment replacing the large trees with stumps and shown the results in Table 3. The results for stumps match those from the larger trees, just with a smaller difference in margins and moment values.

Table 3: Experiments with stumps as base learners. Same setup as in Table 1.

| Data Set | Alg. | Train Err | Test Err | Mean Margin | Moment |
|----------|------|-----------|----------|-------------|--------|
| Forest   | ada  | 0.223     | 0.224    | 0.0754      | 0.987  |
|          | lgb  | 0.217     | 0.218    | 0.0225      | 0.0986 |
| Boone    | ada  | 0.0781    | 0.0817   | 0.138       | 0.975  |
|          | lgb  | 0.0669    | 0.0744   | 0.0422      | 0.239  |
| Higgs    | ada  | 0.309     | 0.31     | 0.059       | 0.986  |
|          | lgb  | 0.301     | 0.302    | 0.0309      | 0.329  |
| Diabetes | ada  | 0.161     | 0.246    | 0.108       | 0.976  |
|          | lgb  | 0.176     | 0.238    | 0.138       | 0.299  |

## 4 Generalization Bound Proof

This section is devoted to the proof of our refined margin based generalization bound for voting classifiers, presented hereafter as Theorem 1. First we recollect some notation. Let $\mathcal{X}$ be some ground set, $\mathcal{D}$ a distribution over $\mathcal{X} \times [-1, 1]$, $\mathcal{H} \subseteq \mathcal{X} \to [-1, 1]$, and $\mathcal{C} = \mathcal{C}(\mathcal{H})$ be the convex hull of $\mathcal{H}$. Fix a voting classifier $f$, then there exists a sequence $\langle \alpha_h \rangle_{h \in \mathcal{H}} \in \mathbb{R}_+^{\mathcal{H}}$ such that $\sum_{h \in \mathcal{H}} \alpha_h = 1$ and $f = \sum_{h \in \mathcal{H}} \alpha_h \cdot h$. Thus $f$ implicitly defines a distribution $\mathcal{Q} = \mathcal{Q}(f)$ over $\mathcal{H}$, where $\Pr_{h \sim \mathcal{Q}}[h = h'] = \alpha_{h'}$ for all $h' \in \mathcal{H}$. Finally, let $\Delta : \mathcal{X} \times \mathcal{H} \to \mathbb{R}$ be defined by $\Delta(x, h) := |f(x) - h(x)|$ for every $x \in \mathcal{X}$, $h \in \mathcal{H}$. We show the following.

**Theorem 1.** *Let $\mathcal{D}$ be a distribution over $\mathcal{X} \times \{-1, 1\}$ where $\mathcal{X}$ is some ground set, and let $\mathcal{H} \subseteq \mathcal{X} \to [-1, 1]$. For every $\delta > 0$, it holds with probability at least $1 - \delta$ over a set of $m$ samples $S \sim \mathcal{D}^m$, that for every voting classifier $f \in \mathcal{C}(\mathcal{H})$ and every margin $\theta > 0$, we have*

$$\mathcal{L}_{\mathcal{D}}(f) \leq \mathcal{L}_S^\theta(f) + O\left( \frac{N \lg |\mathcal{H}| + \lg(1/\delta)}{m} + \sqrt{\frac{N \lg |\mathcal{H}| + \lg(1/\delta)}{m} \mathcal{L}_S^\theta(f)} \right), \quad (6)$$

*where* $N = O\left( \max\{ \theta^{-2} \cdot \left( \mathbb{E}_{(x,y) \sim S} \left[ \mathbb{E}_{h \sim \mathcal{Q}(f)} \left[ \Delta(x, h)^2 \right]^{(\lg(16m))/2} \right] \right)^{2/(\lg(16m))}, \theta^{-1} \} \lg m \right).$

Denote by $\mathcal{E} = \mathcal{E}(\delta)$ the event that for every voting classifier $f$ and every margin $\theta > 0$, the bound in (6) holds with $N$ as defined in Theorem 1. In these notations we prove that $\Pr_{S \sim \mathcal{D}^m}[\mathcal{E}] \geq 1 - \delta$.

**Proof overview.** Inspired by techniques presented by Schapire *et al.* [14] and employed by Gao and Zhou [6], our proof incorporates a discretization of the set of all voting classifiers over $\mathcal{H}$ to a discrete *net* of classifiers, such that, loosely speaking, every voting classifier over $\mathcal{H}$ can be approximated by a classifier that belongs to the net, and in addition, the size of the net is not too big, and thus union bounding over the net yields the desired probability bounds. Thus, intuitively speaking, by randomly rounding every voting classifier $f$ to the net we get an upper bound on the out of sample error for $f$. More specifically, $N \in \mathbb{N}^+$ be some positive integer. We define a net $\mathcal{C}_N$ of voting classifier

by $\mathcal{C}_N := \left\{ \frac{1}{N} \sum_{j \in [N]} h_j : \langle h_j \rangle_{j \in [N]} \in \mathcal{H}^N \right\}$. For every voting classifier $f$ over $\mathcal{H}$, we then give a randomized rounding scheme that essentially associates a random net element $g \in \mathcal{C}_N$ with $f$, and show that with high probability the out of sample error with respect to $g$ well-approximates that of $f$. By choosing $N$ carefully and union bounding over $\mathcal{C}_N$ we get an upper bound on the out of sample error for all voting classifiers $f$. The crux of the proof lies in carefully choosing the size of the net, namely $N$. Loosely speaking, the net size $N$ has to be large enough, so that the net is rich enough to approximate every voting classifier well, but on the other hand small enough, so that union bounding over the net does not incur too large a cost for the probability bound. By subtly choosing $N$ and proving refined bounds on the rounding scheme we get the bound in Theorem 1.

Formally we define for every $N \in \mathbb{N}^+$, the event $\mathcal{E}_N$ to be the set of all samples $S \in (\mathcal{X} \times \{-1, 1\})^m$ satisfying that for all voting classifiers $g \in C_N$ and integer $\ell \in [0, N]$ it holds that

$$\mathcal{L}_{\mathcal{D}}^{\ell/N}(g) \le \mathcal{L}_S^{\ell/N}(g) + \frac{8 \ln(4\delta^{-1} N(N+1)^2 |\mathcal{H}|^N)}{m} + 4\sqrt{\frac{\ln(4N(N+1)^2|\mathcal{H}|^N/\delta)}{m} \mathcal{L}_S^{\ell/N}} ; \quad (7)$$

and

$$\Pr_{(x,y) \sim \mathcal{D}} [ \, |f(x) - g(x)| > \ell/N \, ] \le 2 \Pr_{(x,y) \sim S} [ \, |f(x) - g(x)| > \ell/N \, ] + \frac{8 \ln(4\delta^{-1} N(N+1)^2 |\mathcal{H}|^N)}{m} . \quad (8)$$

Intuitively speaking, for $S \in \mathcal{E}_N$, the first bound ensures a good generalization bound for every voting classifier $g$ in the net, whereas the second bound shows that $g$ approximates $f$ over $\mathcal{D}$ almost as well as it approximates $f$ over $S$. In turn these two bounds imply that the behavior of $f, g$ over $S$ predicts their behavior over $\mathcal{D}$. As $\sum_{N=1}^{\infty} \frac{1}{N(N+1)} = 1$, the following lemma implies Theorem 1 by applying a union bound.

**Lemma 2.** *For every $N$ we have* $\Pr_{S \sim \mathcal{D}^m}[\mathcal{E}_N] \ge 1 - \frac{\delta}{N(N+1)}$, *and moreover,* $\bigcap_{N \in \mathbb{N}^+} \mathcal{E}_N \subseteq \mathcal{E}$.

The proof of the lemma is quite involved technically, and most of the proof is thus deferred to the full version of the paper . Our main novelty lies in showing that for our choice of $N = N(f, \theta)$, for every sample set $S \in \mathrm{supp}(\mathcal{D}^m)$, with very high probability over the choice of a point $x \in \mathcal{X}$ and a net-classifier $g \in \mathcal{C}_N$, $g$ approximates $f$. In turn, this implies that if $S \in \bigcap_{N \in \mathbb{N}^+} \mathcal{E}_N$, then for every voting classifier $f$ and $\theta > 0$, $f$ is well-approximated by a randomized rounding to the net $\mathcal{C}_N$. Formally we show the following for every $f$ and $\theta$.

**Lemma 3.** $\Pr_{\substack{(x,y) \sim S \\ g \sim \mathcal{Q}^N}} [ \, \Delta(x, g) > 49\theta/100 ] \le \frac{1}{m^2}$, *where*

$$N = N(f, \theta) := \lg(16m) \cdot \max\{256\theta^{-1} \|\Delta(x, h)\|_{\lg(16m)}, 100/\theta ,$$

$$128e\theta^{-2} \cdot \left( \mathbb{E}_{(x,y) \sim S} \left[ \mathbb{E}_{h \sim \mathcal{Q}} \left[ \Delta(x, h)^2 \right]^{(\lg(16m))/2} \right] \right)^{2/(\lg(16m))} \} .$$

*Proof.* Let $Z = \Delta(x, g)$, then for every integer $r \ge 1$ we conclude from Markov's inequality that

$$\Pr_{\substack{(x,y) \sim S \\ g \sim \mathcal{Q}^N}} [ \, Z > 49\theta/100 ] = \Pr_{\substack{(x,y) \sim \mathcal{D} \\ g \sim \mathcal{Q}^N}} [Z^r > (49\theta/100)^r] \le \left( \frac{100}{49\theta} \right)^r \|Z\|_r^r . \quad (9)$$

It is therefore enough to show $\|Z\|_r^r \le \left( \frac{49\theta}{100} \right)^r m^{-2}$ for some positive integer $r \ge 1$. Let $r = 2 \cdot \lceil \lg(4m)/2 \rceil$, then $r$ is an even integer, satisfying $\lg(4m) = 2 \lg(4m)/2 \le r \le \lg(4m) + 2 \le N$. Since $r$ is even, then for $g = \frac{1}{N} \sum_{j \in [N]} h_j$ we get that

$$Z^r = Z(x, g)^r = \left( \frac{1}{N} \sum_{j \in [N]} (f(x) - h_j(x)) \right)^r = \frac{1}{N^r} \sum_{T = (j_i)_{i \in [r]} \in [N]^r} \prod_{i \in [r]} (f(x) - h_{j_i}(x)) .$$

For every $T = (j_i)_{i \in [r]} \in [N]^r$ let $D(T) := \{j \in [N] : \exists i \in [r].j_i = j\}$ be the set of distinct indices occurring in $T$, and for every $j \in [N]$, let $c_T(j) := |\{i \in [r] : j_i = j\}|$ be the number of times $j$ occurs in $T$. As $h_1, \ldots, h_N$ are chosen independently, we get that

$$\mathbb{E}_{(h_k)_{k \in [N]} \sim \mathcal{Q}^N}[Z^r] = \frac{1}{N^r} \sum_{T \in [N]^r} \prod_{j \in D(T)} \mathbb{E}_{(h_k)_{k \in [N]} \sim \mathcal{Q}^N} \left[ (f(x) - h_j(x))^{c_T(j)} \right] .$$

Let $T \in [N]^r$, and assume that for some $j \in D(T)$ we have $c_T(j) = 1$, then

$$\mathbb{E}_{(h_k)_{k \in [N]} \sim \mathcal{Q}^N} \left[ (f(x) - h_j(x))^{c_T(j)} \right] = \mathbb{E}_{h \sim \mathcal{Q}} [f(x) - h(x)] = f(x) - \mathbb{E}_{h \sim \mathcal{Q}} [h(x)] = f(x) - \sum_{h \in \mathcal{H}} \alpha_h h(x) = 0 \,,$$

Denote $\mathcal{T} := \{T \in [N]^r : \forall j \in D(T). \ c_T(j) > 1\}$, then

$$\mathbb{E}_{(h_k)_{k \in [N]} \sim \mathcal{Q}^N} [Z^r] = \frac{1}{N^r} \sum_{T \in \mathcal{T}} \prod_{j \in D(T)} \mathbb{E}_{h \sim \mathcal{Q}} \left[ \Delta(x,h)^{c_T(j)} \right] \,. \tag{10}$$

By Lyapunov's Theorem (see, e.g. [12]), $\mathbb{E}_{h \sim \mathcal{Q}}[\Delta(x,h)^\xi]$ is logarithmic convex for $\xi \in [1, +\infty)$, and as $c_T(j) \geq 2$ for all $j \in D(T)$ we get that

$$\prod_{j \in D(T)} \mathbb{E}_{h \sim \mathcal{Q}} \left[ \Delta(x,h)^{c_T(j)} \right] \leq \mathbb{E}_{h \sim \mathcal{Q}} \left[ \Delta(x,h)^2 \right]^{|D(T)|-1} \mathbb{E}_{h \sim \mathcal{Q}} \left[ \Delta(x,h)^{r-2|D(T)|+2} \right] \,.$$

Plugging into (10) we get that

$$\mathbb{E}_{(h_k)_{k \in [N]} \sim \mathcal{Q}^N} [Z^r] \leq \frac{1}{N^r} \sum_{T \in \mathcal{T}} \mathbb{E}_{h \sim \mathcal{Q}} \left[ \Delta(x,h)^2 \right]^{|D(T)|-1} \mathbb{E}_{h \sim \mathcal{Q}} \left[ \Delta(x,h)^{r-2|D(T)|+2} \right] \,. \tag{11}$$

For every $d \in \mathbb{N}$ denote $\mathcal{T}_d := \{T \in \mathcal{T} : |D(T)| = d\}$. Since for every $T \in \mathcal{T}$ and every $j \in D(T)$, we know that $c_T(j) \geq 2$, then for every $d > r/2$ we get that $\mathcal{T}_d = \emptyset$. Therefore $\mathcal{T} = \bigcup_{d \in [r/2]} |\mathcal{T}_d|$. Moreover, for every $d \in [r/2]$ and every $T \in \mathcal{T}_d$, we have

$$\mathbb{E}_{h \sim \mathcal{Q}} \left[ |h(x)|^2 \right]^{|D(T)|-1} \mathbb{E}_{h \sim \mathcal{Q}} \left[ |h(x)|^{r-2|D(T)|+2} \right] = \mathbb{E}_{h \sim \mathcal{Q}} \left[ |h(x)|^2 \right]^{d-1} \mathbb{E}_{h \sim \mathcal{Q}} \left[ |h(x)|^{r-2d+2} \right] \,.$$

We therefore refine (11) to get

$$\mathbb{E}_{(h_k)_{k \in [N]} \sim \mathcal{Q}^N} [Z^r] \leq \frac{1}{N^r} \sum_{d \in [r/2]} |\mathcal{T}_d| \, \mathbb{E}_{h \sim \mathcal{Q}} \left[ \Delta(x,h)^2 \right]^{d-1} \mathbb{E}_{h \sim \mathcal{Q}} \left[ \Delta(x,h)^{r-2d+2} \right] \,. \tag{12}$$

**Claim 4.** *For every $d \in [r/2]$, $|\mathcal{T}_d| \leq r^r \sqrt{2e\pi r} \left( \frac{Ne}{r} \right)^d$.*

*Proof.* Fix some $d \in [r/2]$. There are at most $\binom{N}{d}$ ways to choose a subset $Y \subseteq [N]$ such that $|Y| = d$. Once such a set $Y$ is fixed, there are at most $\binom{d+(r-2d)-1}{r-2d}$ solution to the equation $\sum_{j \in Y} y_j = r$ under the constraint that $y_j \in \mathbb{N} \setminus \{0,1\}$ for all $j \in Y$. Moreover, once $\{y_j\}_{j \in Y}$ is fixed, there are $r! \cdot \prod_{j \in Y} (y_j!)^{-1}$ ways to form a sequence $T$ satisfying that $D(T) = Y$, $c_T(j) = y_j$ for all $j \in Y$ and $c_T(j) = 0$ otherwise. Note that $\prod_{j \in Y} (y_j!) \geq ((r/d)!)^d$ for every choice of $\{y_j\}_{j \in Y}$, and therefore

$$|\mathcal{T}_d| \leq \binom{N}{d} \cdot \binom{r-d-1}{r-2d} \cdot \frac{r!}{((r/d)!)^d} \leq \sqrt{2e\pi r} (Ne)^d \cdot r^{r-d} \leq r^r \sqrt{2e\pi r} \left( \frac{Ne}{r} \right)^d$$

$\square$

Plugging into (12) we conclude that

$$\mathbb{E}_{(h_k)_{k \in [N]} \sim \mathcal{Q}^N} [Z^r] \leq \sqrt{2e\pi r} \left( \frac{r}{N} \right)^r \sum_{d \in [r/2]} \left( \frac{Ne}{r} \right)^d \mathbb{E}_{h \sim \mathcal{Q}} \left[ \Delta(x,h)^2 \right]^{d-1} \mathbb{E}_{h \sim \mathcal{Q}} \left[ \Delta(x,h)^{r-2d+2} \right]$$

As $\left( \frac{Ne}{r} \right)^\xi, \mathbb{E}_{h \sim \mathcal{Q}} \left[ \Delta(x,h)^2 \right]^{\xi-1}, \mathbb{E}_{h \sim \mathcal{Q}} \left[ \Delta(x,h)^{r-2\xi+2} \right]$ are all logarithmic convex for $\xi \in [1, r/2]$, their product is also logarithmic convex over that range, and thus gets its maximum on either 1 or $r/2$. Concluding we get that

$$\mathbb{E}_{(h_k)_{k \in [N]} \sim \mathcal{Q}^N} [Z^r] \leq \frac{r}{2} \cdot \sqrt{2e\pi r} \left( \frac{r}{N} \right)^r \left( \left( \frac{Ne}{r} \right) \mathbb{E}_{h \sim \mathcal{Q}} [\Delta(x,h)^r] + \left( \frac{Ne}{r} \right)^{r/2} \mathbb{E}_{h \sim \mathcal{Q}} \left[ \Delta(x,h)^2 \right]^{r/2} \right) \,.$$

Taking the expectation over $(x,y) \sim \mathcal{D}$ gives

$$\|Z\|_r^r \leq \frac{r}{2} \sqrt{2e\pi r} \left( \frac{r}{N} \right)^r \left( \left( \frac{Ne}{r} \right) \|\Delta(x,h)\|_r^r + \left( \frac{Ne}{r} \right)^{r/2} \mathbb{E}_{(x,y) \sim \mathcal{D}} \left[ \mathbb{E}_{h \sim \mathcal{Q}} \left[ \Delta(x,h)^2 \right]^{r/2} \right] \right) \tag{13}$$

To finish the proof of Lemma 3, we show that our bound on $N$ implies that $\|Z\|_r^r \leq \left( \frac{49\theta}{100} \right)^r m^{-2}$. This part is quite technical, and is thus deferred to the full version of the paper. $\square$

## Statement of potential broader impact

In this work, we have empirically shown that gradient boosters produce voting classifiers where many base learners make predictions of small magnitude. We then used this observation to prove stronger generalization bounds that better explain the practical performance of gradient boosters. We hope and believe that our findings may not only advance our theoretical understanding of boosting algorithms, but potentially also lead to algorithms with better accuracy by using regularization inspired by our new generalization bound or more directly optimizing it.

## Acknowledgments and Disclosure of Funding

Kasper Green Larsen is supported by DFF Sapere Aude Grant 9064-00068B, a Villum Young Investigator Grant and an AUFF Starting Grant. Lior Kamma us supported by a Villum Young Investigator Grant. Allan Grønlund is supported by Innovation Fund Denmark project DABAI IFD-5153-00004B.

## Footnotes

*All authors contributed equally, and are presented in alphabetical order.

†Department of Computer Science, Aarhus University, {jallan,lior.kamma,larsen}@cs.au.dk

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
