[Supplementary Material]

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

 \left(\frac{Ne}{d}\right)^d\cdot 2^{r-d}\cdot\frac{\sqrt{2e\pi r}(r/e)^r}{(\sqrt{2\pi(r/d)}(r/(ed))^{r/d})^d}$$
$$\leq \sqrt{2e\pi r}\,(Ne)^d\cdot r^{r-d} \leq r^r\sqrt{2e\pi r}\left(\frac{Ne}{r}\right)^d$$

$\square$

Plugging into (10) we conclude that

$$\mathop{\mathbb{E}}_{(h_k)_{k\in[N]}\sim\mathcal{Q}^N}[Z^r] \le \frac{1}{N^r}\sum_{d\in[r/2]} r^r\sqrt{2e\pi r}\left(\frac{Ne}{r}\right)^d \mathop{\mathbb{E}}_{h\sim\mathcal{Q}}\left[\Delta(x,h)^2\right]^{d-1}\mathop{\mathbb{E}}_{h\sim\mathcal{Q}}\left[\Delta(x,h)^{r-2d+2}\right]$$

$$= \sqrt{2e\pi r}\left(\frac{r}{N}\right)^r\sum_{d\in[r/2]}\left(\frac{Ne}{r}\right)^d \mathop{\mathbb{E}}_{h\sim\mathcal{Q}}\left[\Delta(x,h)^2\right]^{d-1}\mathop{\mathbb{E}}_{h\sim\mathcal{Q}}\left[\Delta(x,h)^{r-2d+2}\right]$$

As $\left(\frac{Ne}{r}\right)^\xi,\mathbb{E}_{h\sim\mathcal{Q}}\left[\Delta(x,h)^2\right]^{\xi-1},\mathbb{E}_{h\sim\mathcal{Q}}\left[\Delta(x,h)^{r-2\xi+2}\right]$ are all logarithmic convex for $\xi\in[1,r/2]$, their product is also logarithmic convex over that range, and thus gets its maximum on either $1$ or $r/2$. Concluding we get that

$$\mathop{\mathbb{E}}_{(h_k)_{k\in[N]}\sim\mathcal{Q}^N}[Z^r] \le \frac{r}{2}\cdot\sqrt{2e\pi r}\left(\frac{r}{N}\right)^r\left(\left(\frac{Ne}{r}\right)\mathop{\mathbb{E}}_{h\sim\mathcal{Q}}[\Delta(x,h)^r] + \left(\frac{Ne}{r}\right)^{r/2}\mathop{\mathbb{E}}_{h\sim\mathcal{Q}}\left[\Delta(x,h)^2\right]^{r/2}\right)\ .$$

Taking the expectation over $(x,y)\sim\mathcal{D}$ gives

$$\|Z\|_r^r \le \frac{r}{2}\sqrt{2e\pi r}\left(\frac{r}{N}\right)^r\left(\left(\frac{Ne}{r}\right)\|\Delta(x,h)\|_r^r + \left(\frac{Ne}{r}\right)^{r/2}\mathop{\mathbb{E}}_{(x,y)\sim\mathcal{D}}\left[\mathop{\mathbb{E}}_{h\sim\mathcal{Q}}\left[\Delta(x,h)^2\right]^{r/2}\right]\right)$$
(11)

To finish the proof of Lemma 3, we show that our bound on $N$ implies that $\|Z\|_r^r\le\left(\frac{49\theta}{100}\right)^r m^{-2}$. Denote

$$\Psi_1 = \frac{r}{2}\cdot\sqrt{2e\pi r}\left(\frac{r}{N}\right)^r\cdot\left(\frac{Ne}{r}\right)\|\Delta(x,h)\|_r^r = \frac{r}{2}\cdot\sqrt{2e\pi r}\left(\frac{r\|\Delta(x,h)\|_r}{N}\right)^r\cdot\left(\frac{Ne}{r}\right)$$

$$\Psi_2 = \frac{r}{2}\cdot\sqrt{2e\pi r}\left(\frac{r}{N}\right)^r\left(\left(\frac{Ne}{r}\right)^{r/2}\mathop{\mathbb{E}}_{(x,y)\sim\mathcal{D}}\left[\mathop{\mathbb{E}}_{h\sim\mathcal{Q}}\left[\Delta(x,h)^2\right]^{r/2}\right]\right)$$

Plugging into (11) we get that $\|Z\|_r^r\le\Psi_1+\Psi_2$.

We will show that $\max\{\Psi_1,\Psi_2\}\le\left(\frac{49\theta}{100}\right)^r\cdot\frac{1}{2m^2}$, which proves the claim. To bound $\Psi_1$, note first that $\Psi_1$ decreases as a function of $N$ (since $r\ge2$). Since $N\ge256\theta^{-1}\lg(16m)\cdot\|\Delta(x,h)\|_{\lg(16m)}$ we get that

$$\Psi_1\le\frac{r}{2}\cdot\sqrt{2e\pi r}\left(\frac{r\cdot\|\Delta(x,h)\|_r}{256\theta^{-1}\lg(16m)\cdot\|\Delta(x,h)\|_{\lg(16m)}}\right)^r\cdot\left(\frac{256\theta^{-1}\lg(16m)\cdot\|\Delta(x,h)\|_{\lg(16m)}\cdot e}{r}\right)$$

Since $r<\lg(16m)$, and by monotonicity of norms, $\|\Delta(x,h)\|_r\le\|\Delta(x,h)\|_{\lg(16m)}\le2$, where the last inequality is due to the fact that $|f(x)-h(x)|\le2$ for all $h\in\mathcal{H}$, $x\in\mathcal{X}$. Moreover, $\lg(4m)\le r\le\lg(16m)\le2(\lg(4m))$, therefore

$$\Psi_1\le\frac{r}{2}\cdot\sqrt{2e\pi r}\left(\frac{\theta}{256}\right)^r\cdot1024e\theta^{-1}$$

$$\le\left(\frac{49\theta}{100}\right)^r\cdot3r^{3/2}125^{-r}\cdot\left(1024e\theta^{-1}\right)\le\left(\frac{49\theta}{100}\right)^r\cdot3r^{3/2}64^{-\lg m}125^{-2}\cdot\left(1024e\theta^{-1}\right)$$

$$\le\left(\frac{49\theta}{100}\right)^r\cdot\frac{1}{5}\lg^{3/2}(4m)\cdot m^{-6}\theta^{-1}\le\left(\frac{49\theta}{100}\right)^r\cdot\frac{1}{2m^2}\cdot\frac{1}{2}(\lg(4m)/m)^{3/2}(m^{5/2}\theta)^{-1}$$

For large enough $m$, we have that $\lg(4m)/m\le5/8$, and therefore $(\lg(4m)/m)^{3/2}\le1/2$. Since $\theta\ge1/m$ we get that $\Psi_1\le\left(\frac{49\theta}{100}\right)^r\cdot\frac{1}{2m^2}$. We now turn to bound $\Psi_2$. Recall that $N\ge128e\theta^{-2}\lg(16m)\cdot\left(\mathbb{E}_{(x,y)\sim\mathcal{D}}\left[\mathbb{E}_{h\sim\mathcal{Q}}\left[\Delta(x,h)^2\right]^{\lg(16m)/2}\right]\right)^{2/\lg(16m)}$, and therefore

$$\Psi_2\le3r^{3/2}\left(\frac{er\,\mathbb{E}_{(x,y)\sim\mathcal{D}}\left[\mathbb{E}_{h\sim\mathcal{Q}}\left[\Delta(x,h)^2\right]^{r/2}\right]^{2/r}}{128e\theta^{-2}\lg(16m)\left(\mathbb{E}_{(x,y)\sim\mathcal{D}}\left[\mathbb{E}_{h\sim\mathcal{Q}}\left[\Delta(x,h)^2\right]^{\lg(16m)/2}\right]\right)^{2/\lg(16m)}}\right)^{r/2}$$

$$\le\left(\frac{49\theta}{100}\right)^r\cdot3r^{3/2}\left(\frac{r\,\mathbb{E}_{(x,y)\sim\mathcal{D}}\left[\mathbb{E}_{h\sim\mathcal{Q}}\left[\Delta(x,h)^2\right]^{r/2}\right]^{2/r}}{30\lg(16m)\left(\mathbb{E}_{(x,y)\sim\mathcal{D}}\left[\mathbb{E}_{h\sim\mathcal{Q}}\left[\Delta(x,h)^2\right]^{\lg(16m)/2}\right]\right)^{2/\lg(16m)}}\right)^{r/2}$$

Since $r < \log(16m)$, and by monotonicity of norms of random variables, we get that

$$\mathop{\mathbb{E}}_{(x,y)\sim\mathcal{D}}\left[\mathop{\mathbb{E}}_{h\sim\mathcal{Q}}\left[\Delta(x,h)^2\right]^{r/2}\right]^{2/r} \leq \mathop{\mathbb{E}}_{(x,y)\sim\mathcal{D}}\left[\mathop{\mathbb{E}}_{h\sim\mathcal{Q}}\left[\Delta(x,h)^2\right]^{\log(16m)/2}\right]^{2/\log(16m)}.$$

Therefore

$$\Psi_2 \leq \left(\frac{49\theta}{100}\right)^r \cdot 3r^{3/2}(30)^{-r/2} \leq \left(\frac{49\theta}{100}\right)^r \cdot 3r^{3/2}(30)^{-(\lg m)/2-1} \leq \left(\frac{49\theta}{100}\right)^r \cdot \frac{1}{2m^2} \cdot \frac{1}{5}r^{3/2}m^{-2/5}$$

Similarly to before, for large enough $m$, $\lg^{3/2}(4m)\cdot m^{-2/5} \leq 5$, and therefore we conclude that $\Psi_2 \leq \left(\frac{49\theta}{100}\right)^r \cdot \frac{1}{2m^2}$, which completes the proof of the lemma. $\qquad\square$

## 5 Generalization lower bound

In this section we state and prove our new generalization lower bound, presented as Theorem 5.

**Theorem 5.** *For every large enough integer $N$, every $\theta \in (1/N, 1/40)$, $\tau \in [0,1]$ and every $\left(\theta^{-2}\ln N\right)^{1+\Omega(1)} \leq m \leq 2^{N^{O(1)}}$, if $\frac{\ln N \ln m}{m\theta^2} \leq \tau \leq 1$, then there exist a set $\mathcal{X}$, a hypothesis set $\mathcal{H}$ over $\mathcal{X}$ and a distribution $\mathcal{D}$ over $\mathcal{X} \times \{-1,1\}$ such that $\ln|\mathcal{H}| = \Theta(\ln N)$ and with probability at least $1/100$ over the choice of samples $S \sim \mathcal{D}^m$ there exists a voting classifier $f_S \in C(\mathcal{H})$ such that*

1. *$\mathcal{L}_S^\theta(f_S) \leq \tau$; and*

2. *$\mathcal{L}_\mathcal{D}(f_S) \geq \mathcal{L}_S^\theta(f_S) + \Omega\left(\frac{\ln|\mathcal{H}|\ln m}{m\theta^2} + \sqrt{\tau\ln(\tau^{-1})\cdot\frac{\ln|\mathcal{H}|}{m\theta^2}}\right)$.*

Our proof is inspired by the constructions in [9, 8] and makes use of the following lemma, whose proof can be found in [8].

**Lemma 6.** *For every $\theta \in (0, 1/40)$, $\delta \in (0,1)$ and integers $d \leq u$, there exists a distribution $\mu = \mu(u,d,\theta,\delta)$ over hypothesis sets $\mathcal{H} \subset \mathcal{X} \to \{-1,1\}$, where $\mathcal{X}$ is a set of size $u$, such that the following holds for $N = \Theta\left(\theta^{-2}\ln d\ln(\theta^{-2}d\delta^{-1})e^{\Theta(\theta^2 d)}\right)$.*

1. *For all $\mathcal{H} \in \mathrm{supp}(\mu)$, we have $|\mathcal{H}| = N$; and*

2. *For every labeling $\ell \in \{-1,+1\}^u$, if no more than $d$ points $x \in \mathcal{X}$ satisfy $\ell(x) = -1$, then*

$$\Pr_{\mathcal{H}\sim\mu}\left[\exists f \in \mathcal{C}(\mathcal{H}) : \forall x \in \mathcal{X}. \ \ell(x)f(x) \geq \theta\right] \geq 1-\delta,$$

We start by describing the outlines of the proofs. To this end fix some integer $N$, and fix $\theta \in (1/N, 1/40)$. Let $u$ be an integer, and let $\mathcal{X} = \{\xi_1, \ldots, \xi_u\}$ be some set with $u$ elements. The distribution $\mathcal{D}$ over $\mathcal{X} \times \{-1,1\}$, is simply the uniform distribution over $\mathcal{X} \times \{1\}$. That is for every $i \in [u]$ and $y \in \{-1,1\}$, $\Pr_\mathcal{D}[(\xi_i, y)] = \frac{1+y}{2u}$. The following claim is straightforward.

**Claim 7.** *For every $f : \mathcal{X} \to \mathbb{R}$ we have $\Pr_{(x,y)\sim\mathcal{D}}[yf(x) < 0] = \frac{1}{u}\sum_{i\in[u]}\mathbb{1}_{f(\xi_i)<0}$.*

We will show that with some constant probability over a random choice $S \sim \mathcal{D}^m$, an adversarial voting classifier has a high generalization probability. We additionally show existence of a hypothesis set $\hat{\mathcal{H}}$ such that with very high (constant) probability over a random choice of $\ell \in \{-1,1\}^u$, $C(\hat{\mathcal{H}})$ contains a voting classifier that attains high margins with $\ell$ over the entire set $\mathcal{X}$. Finally, we conclude that with positive probability over a random choice of $S \sim \mathcal{D}^m$ both properties are satisfied.

To prove existence of a "rich" yet small enough hypothesis set $\hat{\mathcal{H}}$ we apply Lemma 6 together with Yao's minimax principle. In order to ensure that the hypothesis sets constructed using Lemma 6 is small enough, and specifically has size $N^{O(1)}$, we need to focus our attention on sparse labelings $\ell \in \{-1,1\}^u$ only. That is, the labelings cannot contain more than $\frac{\ln N}{\theta^2}$ entries equal to $-1$. To this end we will focus on $d$-sparse vectors. More formally, we define a set of labelings of interest $\mathcal{L}(u,d)$ as follows.

$$\mathcal{L}(u,d) := \{\ell \in \{-1,1\}^u : |\{i \in [u] : \ell_i = -1\}| \leq d\}. \tag{12}$$

We next show that there exists a small enough (with respect to $N$) hypothesis set $\hat{\mathcal{H}}$ that is rich enough. That is, with high probability over $\ell \in \mathcal{L}(u, d)$, there exists a voting classifier $f \in C(\hat{\mathcal{H}})$ that attains high minimum margin with $\ell$ over the entire set $\mathcal{X}$. Note that the following result, similarly to Lemma 6 does not depend on the size of $\mathcal{X}$, but only on the sparsity of the labelings in question.

**Claim 8.** *If $u \leq 2^{N^{O(1)}}$ and $d \leq \frac{\ln N}{\theta^2}$ then there exists a hypothesis set $\hat{\mathcal{H}}$ such that $\ln |\hat{\mathcal{H}}| = \Theta\left(\ln N\right)$ and*

$$\Pr_{\ell \in_R \mathcal{L}(u, d)}[\exists f \in C(\hat{\mathcal{H}}) : \forall i \in [u]. \ \ell_i f(\xi_i) \geq \theta] \geq 1 - 1/N \ .$$

*Proof.* Let $\mu = \mu(u, d, \theta, 1/N)$, be the distribution whose existence is guaranteed in Lemma 6. Then for every labeling $\ell \in \mathcal{L}(u, d)$, with probability at least $99/100$ over $\mathcal{H} \sim \mu$, there exists a voting classifier $f \in C(\mathcal{H})$ that has minimal margin of $\theta$. That is, for every $i \in [u]$, $\ell_i f(\xi_i) \geq \theta$. By Yao's minimax principle, there exists a hypothesis set $\hat{\mathcal{H}} \in \text{supp}(\mu)$ such that

$$\Pr_{\ell \in_R \mathcal{L}(u, d)}[\exists f \in C(\hat{\mathcal{H}}) : \forall i \in [u]. \ \ell_i f(x_i) \geq \theta] \geq 1 - 1/N \ .$$

Moreover, since $\hat{\mathcal{H}} \in \text{supp}(\mu)$, then $|\hat{\mathcal{H}}| = \Theta\left(\theta^{-2} \ln u \cdot \ln(N\theta^{-2} \ln u) \cdot e^{\Theta(\theta^2 d)}\right)$. Since $\theta \geq 1/N$, $\ln u \leq N^{O(1)}$, and $d \leq \frac{\ln N}{\theta^2}$, and thus $e^{\theta^2 d} = N$ we get that there exists some universal constant $C > 0$ such that $|\hat{\mathcal{H}}| = \Theta(N^C)$, and thus $\ln |\hat{\mathcal{H}}| = \Theta(\ln N)$. $\qquad \square$

Let $u = \frac{\ln N}{16\tau\theta^2}$, and let $d = \frac{\ln N}{16e^{28}\theta^2}$. We next introduce some notation. With every set $T \subseteq [u]$ we associate the classifier $h_T : \mathcal{X} \to \{-1, 1\}$ satisfying that for every $x \in \mathcal{X}$, $h_T(x) = -1$ if and only if $x \in T$. For every $m$-point sample $S \in (\mathcal{X} \times \{1\})^m$ and every $i \in [u]$, let $b_i^S$ be the number of times $\xi_i$ is sampled into $S$. If the set $S$ is clear from context, we simply denote $b_i$. In these notations, $\mathcal{L}_S(h_T) = \frac{1}{m} \sum_{i \in T} b_i^S$ for every $T \subseteq [u]$. Given a sample set $S$ Let $T^* = T^*(S) \subseteq [u]$ be a random set of size $d$ that minimizes $\mathcal{L}_S(h_{T^*(S)}) = \sum_{i \in T^*(S)} b_i^S$. We will show the following.

**Lemma 9.** *With probability at least $1/100$ over the choice of sample $S \sim \mathcal{D}^m$, the following holds.*

1. *There exists a voting classifier $f_S \in C(\hat{\mathcal{H}})$ such that $f_S(\xi_i) h_{T^*(S)}(\xi_i) \geq \theta$ for all $i \in [u]$; and*

2. $\mathcal{L}_S(h_{T^*(S)}) \leq \frac{d}{u}\left(1 - \sqrt{\frac{\ln(u/2d)}{9m/u}}\right)$ .

Note that as $\tau \geq \frac{\ln N \ln m}{m\theta^2}$ we know that $u = \frac{\ln N}{16\tau\theta^2} \leq \frac{m}{16 \ln m}$ and therefore $\frac{\ln(u/2d)}{9m/u} \leq \frac{u \ln(e^{28}/\tau)}{9m} \leq \frac{\ln(e^{28}/\tau)}{144 \ln m} \leq \frac{1}{2}$ for large enough $N$, and therefore the bound in the second part of Lemma 9 is meaningful. We first show that the lemma implies Theorem 5.

*Proof of Theorem 5.* Fix some $\frac{\ln N \ln m}{m\theta^2} \leq \tau \leq 1$. From Lemma 9 with probability $1/100$ over the choice of a sample $S \sim \mathcal{D}^m$ there exists a voting classifier $f_S \in C(\hat{\mathcal{H}})$ such that $f_S(\xi_i) h_{T^*(S)}(\xi_i) \geq \theta$ for all $i \in [u]$ and moreover $\mathcal{L}_S(h_{T^*(S)}) \leq \tau$. Consider $f_S$, and note first that

$$\mathcal{L}_{\mathcal{D}}(f_S) = \frac{1}{u} \sum_{i \in [u]} \mathbb{1}_{f_S(\xi_i) < 0} = \frac{1}{u} \sum_{i \in [u]} \mathbb{1}_{h_{T^*(S)}(\xi_i) < 0} = \frac{|T^*(S)|}{u} = \frac{d}{u} \ .$$

Additionally, since for every $i \in [u]$, $f_S(\xi_i) \leq 0$ if and only if $f_S(\xi_i) \leq \theta$, then

$$\mathcal{L}_S^\theta(f_S) = \mathcal{L}_S(f_S) = \mathcal{L}_S(h_{T^*(S)}) \leq \frac{d}{u}\left(1 - \sqrt{\frac{\ln(u/2d)}{9m/u}}\right) \leq \frac{d}{2u} \leq \tau \ .$$

Summing up we get also that

$$\mathcal{L}_{\mathcal{D}}(f_S) - \mathcal{L}_S^\theta(f_S) \geq \frac{d}{u}\sqrt{\frac{\ln(u/2d)}{9m/u}} = \Omega\left(\tau\sqrt{\frac{u \ln(\tau^{-1})}{m}}\right) = \Omega\left(\sqrt{\frac{\ln N \tau \ln(\tau^{-1})}{m\theta^2}}\right) \ .$$

$\qquad \square$

For the rest of the section we therefore prove Lemma 9. First note that since $\mathcal{D}$ is uniform over $\mathcal{X} \times \{1\}$, and since given $S \sim \mathcal{D}^m$, $T^*$ is sampled uniformly over all subsets $T \in \binom{[u]}{d}$ such that the sum $\sum_{i \in T} b_i^S$ is minimized, we get that for every $T \in \binom{[u]}{d}$, $\Pr_{S \sim \mathcal{D}^m}[T^*(S) = T] = \binom{u}{d}^{-1}$. In other words, for every $h \in \mathcal{L}(u, d)$, $\Pr_{S \sim \mathcal{D}^m}[h_{T^*(S)} = h] = \binom{u}{d}^{-1}$. Therefore $h_{T^*(S)}$ is uniformly distributed over $\mathcal{L}(u, d)$. From claim 8 it follows that for large enough $N$, the probability over the choice of $S \sim \mathcal{D}^m$ that there exists $f_S \in C(\hat{\mathcal{H}})$ such that $f_S(\xi)h_{T^*(S)}(\xi_i) \geq \theta$ for all $i \in [u]$ is at least $99/100$. In order to prove Lemma 9, it is therefore enough to show that with probability at least $1/50$ over the choice of $S \sim \mathcal{D}^m$, $\mathcal{L}_S(h_{T^*(S)}) \leq \frac{d}{u}\left(1 - \sqrt{\frac{\ln(u/2d)}{9m/u}}\right)$. We will show that with probability at least $1/50$ over the choice of $S$ there exist $i_1, \ldots, i_d \in [u]$ such that for every $j \in [d]$, $b_{i_j}^S \leq \frac{m}{u}\left(1 - \sqrt{\frac{\ln(u/2d)}{9m/u}}\right)$. Since $T^*(S)$ minimizes $\sum_{i \in T^*(S)} b_i^S$, it follows that

$$\mathcal{L}_S(h_{T^*(S)}) = \frac{1}{m}\sum_{i \in T^*(S)} b_i^S \leq \frac{1}{m}\sum_{j \in [d]} b_{i_j}^S \leq \frac{d}{u}\left(1 - \sqrt{\frac{\ln(u/2d)}{9m/u}}\right) .$$

To this end, fix some $i \in [u]$. For every $j \in [m]$, let $I_j^S$ be an indicator for the event that the $j$th element selected into $S$ is $(\xi_i, 1)$. Then $b_i^S = \sum_{j \in [m]} I_j^S$, and as $\mathcal{D}$ is uniform, we get that $\mathbb{E}[b_i^S] = \sum_{j \in [m]} \mathbb{E}[I_j^S] = m/u$. We will use the following reverse Chernoff bound and show that with good enough probability, $b_i^S$ is far from its expectation.

**Lemma 10.** *Let $m \in \mathbb{N}^+$ and let $I_1, \ldots, I_m$ be independent indicator random variables with success probability $1/u$. Then for every $\sqrt{3/(m/u)} \leq \delta \leq 1/2$ we have*

$$\Pr\left[\sum_{j \in [m]} I_j \leq (1 - \delta)mp\right] \geq e^{-9m\delta^2/u} .$$

Denote $\delta := \sqrt{\frac{\ln(u/2d)}{9m/u}}$. As we have shown earlier, $\delta \leq 1/2$. Moreover, since $\frac{u}{2d} \geq e^{27}\tau^{-1} \geq e^{27}$, we get that $\delta \geq \sqrt{\frac{27}{9m/u}} = \sqrt{\frac{3}{m/u}}$. We can therefore conclude from Lemma 10 that

$$\Pr[b_i^S \geq (1 - \delta)m/u] \geq e^{-9m\delta^2/u} = e^{-\ln(u/2d)} = \frac{2d}{u} .$$

Let $B_i^S$ be the indicator for the event $b_i^S \geq (1 - \delta)m/u$, then $\mathbb{E}[B_i^S] \geq \frac{2d}{u}$. Finally, let $B^S = \sum_{i \in [u]} B_i^S$, then $\mathbb{E}[B^S] \geq 2d$. We will show that with probability at least $1/8 \geq 1/50$ we have $B^S \geq d$. This implies that there exist $i_1, \ldots, i_d$ such that for every $j \in [d]$, $b_{i_j}^S \leq \frac{m}{u}(1 - \delta) = \frac{m}{u}\left(1 - \sqrt{\frac{\ln(u/2d)}{9m/u}}\right)$. To show $B^S \geq d$ with reasonable probability, we use the Paley-Zigmund inequality.

$$\Pr[B^S \geq d] = \Pr\left[B^S \geq \frac{1}{2}\mathbb{E}[B^S]\right] \geq \frac{\mathbb{E}[B^S]^2}{4\mathbb{E}[(B^S)^2]} .$$

Since $B_1^S, \ldots, B_u^S$ are negatively correlated, we have that $\mathbb{E}[B_i^S B_j^S] \leq \mathbb{E}[B_i^S]\mathbb{E}[B_j^S] = \mathbb{E}[B_1^S]^2$ for every $i, j \in [u]$. Moreover, as $B_1^S, \ldots, B_u^S$ are indicators, $\mathbb{E}[(B_i^S)^2] = \mathbb{E}[B_i^S]$ for all $i \in [u]$. Therefore

$$\mathbb{E}[(B^S)^2] = \sum_{i,j \in [u]} \mathbb{E}[B_i^S B_j^S] \leq (u^2 - u)\mathbb{E}[B_1^S]^2 + u\mathbb{E}[B_i^S]$$
$$\leq u^2\mathbb{E}[B_1^S]^2 + \mathbb{E}[B^S] = \mathbb{E}[B^S]^2 + \mathbb{E}[B^S] \leq 2\mathbb{E}[B^S]^2 ,$$

where the last inequality is due to the fact that $\mathbb{E}[B^S] \geq 2d \geq 1$. We conclude that

$$\Pr[B^S \geq d] \geq \frac{\mathbb{E}[B^S]^2}{4\mathbb{E}[(B^S)^2]} \geq \frac{1}{8} .$$

The proof of the lemma, and therefore of Theorem 5 is now complete.

## Statement of potential broader impact

In this work, we have empirically shown that gradient boosters produce voting classifiers where many base learners make predictions of small magnitude. We then used this observation to prove stronger generalization bounds that better explain the practical performance of gradient boosters. We hope and believe that our findings may not only advance our theoretical understanding of boosting algorithms, but potentially also lead to algorithms with better accuracy by using regularization inspired by our new generalization bound or more directly optimizing it.

## Acknowledgments and Disclosure of Funding

Kasper Green Larsen is supported by DFF Sapere Aude Grant 9064-00068B, a Villum Young Investigator Grant and an AUFF Starting Grant. Lior Kamma us supported by a Villum Young Investigator Grant. Allan Grønlund is supported by Innovation Fund Denmark project DABAI IFD-5153-00004B.

## Footnotes

*All authors contributed equally, and are presented in alphabetical order.

†Department of Computer Science, Aarhus University, {jallan,lior.kamma,larsen}@cs.au.dk

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

## A  Proof of Lemma 2

We start by handling the first part of the lemma, namely that for every $N \in \mathbb{N}^+$, with high probability over $S \sim \mathcal{D}^m$, $S \in \mathcal{E}_N$.

**Claim 11.** *For every $N \in \mathbb{N}^+$, $g \in \mathcal{C}_N$ and $\ell \in [0, N]$, with probability at least $1 - \frac{\delta}{N(N+1)^2|\mathcal{H}|^N}$ over $S \sim \mathcal{D}^m$ we have*

$$\mathcal{L}_{\mathcal{D}}^{\ell/N}(g) \leq \mathcal{L}_S^{\ell/N}(g) + \frac{8\ln(4\delta^{-1}N(N+1)^2|\mathcal{H}|^N)}{m} + 4\sqrt{\frac{\ln(4N(N+1)^2|\mathcal{H}|^N/\delta)}{m}\mathcal{L}_S^{\ell/N}} \; ; \quad (13)$$

*and*

$$\Pr_{(x,y)\sim\mathcal{D}}[\,|f(x) - g(x)| > \ell/N\,] \leq 2\Pr_{(x,y)\sim S}[\,|f(x) - g(x)| > \ell/N\,] + \frac{8\ln(4\delta^{-1}N(N+1)^2|\mathcal{H}|^N)}{m} \; . \quad (14)$$

We draw the reader's attention to the fact that by union bounding over all $g \in \mathcal{C}_N$ and $\ell \in [0, N]$ we get that $\Pr_{S\sim\mathcal{D}^m}[\mathcal{E}_N] \geq 1 - \frac{\delta}{N(N+1)}$ for every $N \in \mathbb{N}^+$, which proves the first part of Lemma 2. The proof of Claim 11 is quite involved technically, and is therefore deferred to the appendix .

*Proof.* First note that if $\mathcal{L}_{\mathcal{D}}^{\ell/N}(g) \leq 8m^{-1}\ln(4\delta^{-1}N(N+1)^2|\mathcal{H}|^N)$ then (13) holds for all $S$, and thus with probability 1 over $S \sim \mathcal{D}^m$. Assume therefore that $\mathcal{L}_{\mathcal{D}}^{\ell/N}(g) > 8m^{-1}\ln(2\delta^{-1}N(N+1)^2|\mathcal{H}|^N)$. Denote $S = \{(x_j, y_j)\}_{j\in[m]}$, then

$$\mathcal{L}_S^{\ell/N}(g) = \Pr_{(x,y)\sim S}[yg(x) \leq \ell/N] = \frac{1}{m}\sum_{j\in[m]}\mathbb{1}_{yg(x_j)\leq\ell/N} \; .$$

Moreover $\mathbb{E}[\mathbb{1}_{yg(x_j)\leq\ell/N}] = \mathcal{L}_{\mathcal{D}}^{\ell/N}(g)$ for all $j \in [m]$, and therefore $\mathbb{E}[\mathcal{L}_S^{\ell/N}(g)] = \mathcal{L}_{\mathcal{D}}^{\ell/N}(g)$. Let $\gamma := \sqrt{\frac{2\ln(4N(N+1)^2|\mathcal{H}|^N/\delta)}{m\mathcal{L}_{\mathcal{D}}^{\ell/N}}}$. Then $\gamma \in (0, 1/2)$, and therefore a Chernoff bound gives the following two inequalities.

$$\Pr_{S\sim\mathcal{D}^m}\left[\mathcal{L}_S^{\ell/N}(g) < (1-\gamma)\mathcal{L}_{\mathcal{D}}^{\ell/N}(g)\right] \leq e^{-\gamma^2 m\mathcal{L}_{\mathcal{D}}^{\ell/N}(g)/2} \leq \frac{\delta}{4N(N+1)^2|\mathcal{H}|^N}$$

$$\Pr_{S\sim\mathcal{D}^m}\left[\mathcal{L}_S^{\ell/N}(g) > 2\mathcal{L}_{\mathcal{D}}^{\ell/N}(g)\right] \leq e^{-m\mathcal{L}_{\mathcal{D}}^{\ell/N}(g)/3} \leq \frac{\delta}{4N(N+1)^2|\mathcal{H}|^N} \; ,$$

where the last inequality follows from the fact that $\mathcal{L}_{\mathcal{D}}^{\ell/N}(g) \geq 8m^{-1}\ln(2\delta^{-1}N(N+1)^2|\mathcal{H}|^N)$. Therefore with probability at least $1 - \delta/(2N(N+1)^2|\mathcal{H}|^N)$ we get that

$$\mathcal{L}_{\mathcal{D}}^{\ell/N}(g) \leq (1-\gamma)^{-1}\mathcal{L}_S^{\ell/N}(g) \leq (1+2\gamma)\mathcal{L}_S^{\ell/N}(g) \leq (1+2\gamma)\mathcal{L}_S^{\ell/N}(g) + \frac{8\ln(2\delta^{-1}N(N+1)^2|\mathcal{H}|^N)}{m} \; , \quad (15)$$

and moreover

$$\gamma = \sqrt{\frac{2\ln(N(N+1)^2|\mathcal{H}|^N/\delta)}{m\mathcal{L}_{\mathcal{D}}^{\ell/N}(g)}} \leq \sqrt{\frac{4\ln(N(N+1)^2|\mathcal{H}|^N/\delta)}{m\mathcal{L}_S^{\ell/N}(g)}} \quad (16)$$

Plugging (16) into (15) and summing up we get

$$\mathcal{L}_{\mathcal{D}}^{\ell/N}(g) \leq \mathcal{L}_S^{\ell/N}(g) + \frac{8\ln(2\delta^{-1}N(N+1)^2|\mathcal{H}|^N)}{m} + 4\sqrt{\frac{\ln(N(N+1)^2|\mathcal{H}|^N/\delta)}{m}\mathcal{L}_S^{\ell/N}(g)} \; .$$

Next note once again that if $\Pr_{(x,y)\sim\mathcal{D}}[\,|f(x)-g(x)|>\ell/N\,]\leq 8m^{-1}\ln(4\delta^{-1}N(N+1)^2|\mathcal{H}|^N)$ then (14) holds for all $S$, and thus with probability 1 over $S\sim\mathcal{D}^m$. Assume therefore that $\Pr_{(x,y)\sim\mathcal{D}}[\,|f(x)-g(x)|>\ell/N\,]>8m^{-1}\ln(4\delta^{-1}N(N+1)^2|\mathcal{H}|^N)$. Similarly to the first part of the proof a Chernoff bound gives the following inequality.

$$\Pr_{S\sim\mathcal{D}^m}\left[\Pr_{(x,y)\sim S}[\,|f(x)-g(x)|>\ell/N\,]>2\Pr_{(x,y)\sim\mathcal{D}}[\,|f(x)-g(x)|>\ell/N\,]\right]$$

$$\leq e^{-m\Pr_{(x,y)\sim\mathcal{D}}[\,|f(x)-g(x)|>\ell/N\,]/3}\leq\frac{\delta}{4N(N+1)^2|\mathcal{H}|^N}\ ,$$

where the last inequality follows from the fact that $\Pr_{(x,y)\sim\mathcal{D}}[\,|f(x)-g(x)|>\ell/N\,]\geq 8m^{-1}\ln(2\delta^{-1}N(N+1)^2|\mathcal{H}|^N)$. Therefore with probability at least $1-\delta/(2N(N+1)^2|\mathcal{H}|^N)$ we get (14). Union bounding we get that with probability with probability at least $1-\delta/(N(N+1)^2|\mathcal{H}|^N)$ over the choice of $S\sim\mathcal{D}^m$ we have both (13) and (14). $\qquad\square$

We turn now to prove the second part of Lemma 2, namely that $\bigcap_{N\in\mathbb{N}^+}\mathcal{E}_N\subseteq\mathcal{E}$. To this end, let $S\in\bigcap_{N\in\mathbb{N}^+}\mathcal{E}_N$. Let $f$ be some voting classifier and let $\theta>0$. As $f$ is a voting classifier, then there exists a sequence $\langle\alpha_h\rangle_{h\in\mathcal{H}}\in\mathbb{R}^{\mathcal{H}}_+$ such that $\sum_{h\in\mathcal{H}}\alpha_h=1$ and $f=\sum_{h\in\mathcal{H}}\alpha_h\cdot h$. Thus $f$ implicitly defines a distribution $\mathcal{Q}=\mathcal{Q}(f)$ over $\mathcal{H}$, where $\Pr_{h\sim\mathcal{Q}}[h=h']=\alpha_{h'}$ for all $h'\in\mathcal{H}$. Recall that $\Delta:\mathcal{X}\times\mathcal{H}\to\mathbb{R}$ is defined by $\Delta(x,h):=|f(x)-h(x)|$ for every $x\in\mathcal{X}$, $h\in\mathcal{H}$.

**Definition 1.** *Let $X$ be a random variable, and let $r\in\mathbb{N}$, then the $r$th moment of $X$ is defined by $\|X\|^r_r:=\mathbb{E}[X^r]$. The $r$th norm of $X$ is defined by $\|X\|_r:=\sqrt[r]{\mathbb{E}[X^r]}$.*

Set hereafter

$$N:=\lg(16m)\cdot\max\{256\theta^{-1}\|\Delta(x,h)\|_{\lg(16m)},100/\theta\ ,$$

$$128e\theta^{-2}\cdot\left(\underset{(x,y)\sim S}{\mathbb{E}}\left[\underset{h\sim\mathcal{Q}}{\mathbb{E}}\left[\Delta(x,h)^2\right]^{(\lg(16m))/2}\right]\right)^{2/(\lg(16m))}\}$$

The product distribution $\mathcal{Q}^N$ defines a distribution over $\mathcal{H}^N$. By identifying an $N$-tuple $h_1,\ldots,h_N\in\mathcal{H}$ with the corresponding classifier $\frac{1}{N}\sum_{j\in[N]}h_j$ we can think of $\mathcal{Q}^N$ also as a distribution over $\mathcal{C}_N$. We first observe that

$$\mathcal{L}_\mathcal{D}(f)\leq\Pr_{(x,y)\sim\mathcal{D},g\sim\mathcal{Q}^N}[yf(x)\leq 0\wedge yg(x)\leq\theta/2]+\Pr_{(x,y)\sim\mathcal{D},g\sim\mathcal{Q}^N}[yf(x)\leq 0\wedge yg(x)>\theta/2]$$

$$\leq\Pr_{(x,y)\sim\mathcal{D},g\sim\mathcal{Q}^N}[yg(x)\leq\theta/2]+\Pr_{(x,y)\sim\mathcal{D},g\sim\mathcal{Q}^N}[\,|f(x)-g(x)|>\theta/2]$$

$$(17)$$

To bound the first summand, let $\ell\in[0,N]$ be the smallest integer such that $\theta/2\leq\ell/N$. Such $\ell$ clearly exists as $\theta\in[0,1]$. Moreover we know that $\theta/2\leq\ell/N\leq\theta/2+1/N\leq 51\theta/100$. Since $S\in\mathcal{E}_N$ we get that

$$\Pr_{\substack{(x,y)\sim\mathcal{D}\\g\sim\mathcal{Q}^N}}[yg(x)\leq\theta/2]\leq\Pr_{\substack{(x,y)\sim\mathcal{D}\\g\sim\mathcal{Q}^N}}[yg(x)\leq\ell/N]=\underset{g\sim\mathcal{Q}^N}{\mathbb{E}}\left[\Pr_{(x,y)\sim\mathcal{D}}[yg(x)\leq\ell/N]\right]$$

$$\leq\underset{g\sim\mathcal{Q}^N}{\mathbb{E}}\left[\Pr_{(x,y)\sim S}[yg(x)\leq\ell/N]+\varepsilon_N(g)\right]\leq\Pr_{\substack{(x,y)\sim S\\g\sim\mathcal{Q}^N}}[yg(x)\leq 51\theta/100]+\underset{g\sim\mathcal{Q}^N}{\mathbb{E}}[\varepsilon_N(g)]\ ,$$

where $\varepsilon_N(g)=\frac{8\ln(2\delta^{-1}N(N+1)^2|\mathcal{H}|^N)}{m}+4\sqrt{\frac{\ln(N(N+1)^2|\mathcal{H}|^N/\delta)}{m}\mathcal{L}^{\ell/N}_S(g)}$. Similarly to (17) we get that

$$\Pr_{\substack{(x,y)\sim S\\g\sim\mathcal{Q}^N}}[yg(x)\leq 51\theta/100]\leq\Pr_{(x,y)\sim S}[yf(x)\leq\theta]+\Pr_{\substack{(x,y)\sim S\\g\sim\mathcal{Q}^N}}[\,|f(x)-g(x)|>49\theta/100]\ ,$$

and therefore

$$\Pr_{\substack{(x,y)\sim\mathcal{D}\\g\sim\mathcal{Q}^N}}[yg(x)\leq\theta/2]\leq\Pr_{(x,y)\sim S}[yf(x)\leq\theta]+\Pr_{\substack{(x,y)\sim S\\g\sim\mathcal{Q}^N}}[\,|f(x)-g(x)|>49\theta/100]+\underset{g\sim\mathcal{Q}^N}{\mathbb{E}}[\varepsilon_N(g)]\ .$$

$$(18)$$

Moreover, since $S \in \mathcal{E}_N$ we get the following bound over the second summand in (17).

$$\Pr_{\substack{(x,y)\sim\mathcal{D}\\g\sim\mathcal{Q}^N}}[\,|f(x) - g(x)| > \theta/2] \leq \Pr_{\substack{(x,y)\sim\mathcal{D}\\g\sim\mathcal{Q}^N}}[\,|f(x) - g(x)| > (\ell - 1)/N]$$

$$\leq 2 \Pr_{\substack{(x,y)\sim S\\g\sim\mathcal{Q}^N}}[\,|f(x) - g(x)| > (\ell - 1)/N] + \frac{8\ln(2\delta^{-1}N(N+1)^2|\mathcal{H}|^N)}{m} \quad (19)$$

$$\leq 2 \Pr_{\substack{(x,y)\sim S\\g\sim\mathcal{Q}^N}}[\,|f(x) - g(x)| > 49\theta/100] + \frac{8\ln(2\delta^{-1}N(N+1)^2|\mathcal{H}|^N)}{m}$$

Plugging (18) and (19) into (17) we get that

$$\mathcal{L}_{\mathcal{D}}(f) \leq \Pr_{(x,y)\sim S}[yf(x) \leq \theta] + 3 \Pr_{\substack{(x,y)\sim S\\g\sim\mathcal{Q}^N}}[\,|f(x) - g(x)| > 49\theta/100]$$

$$+ \frac{16\ln(2\delta^{-1}N(N+1)^2|\mathcal{H}|^N)}{m} + \mathop{\mathbb{E}}_{g\sim\mathcal{Q}^N}\left[\sqrt{\frac{\ln(N(N+1)^2|\mathcal{H}|^N/\delta)}{m}\mathcal{L}_S^{\ell/N}(g)}\right] \quad (20)$$

From Lemma 3 we get that by Jensen's inequality and sub-additivity of square root

$$\mathop{\mathbb{E}}_{g\sim\mathcal{Q}^N}\left[\sqrt{\frac{\ln(N(N+1)^2|\mathcal{H}|^N/\delta)}{m}\mathcal{L}_S^{\ell/N}(g)}\right] \leq \sqrt{\frac{\ln(N(N+1)^2|\mathcal{H}|^N/\delta)}{m}\mathop{\mathbb{E}}_{g\sim\mathcal{Q}^N}\left[\mathcal{L}_S^{51\theta/100}(g)\right]}$$

$$\leq \sqrt{\frac{\ln(N(N+1)^2|\mathcal{H}|^N/\delta)}{m}\left(\mathcal{L}_S^{\theta}(f) + \frac{1}{m^2}\right)}$$

$$\leq \frac{1}{m} + \sqrt{\frac{\ln(N(N+1)^2|\mathcal{H}|^N/\delta)}{m}\mathcal{L}_S^{\theta}(f)}\,, \quad (21)$$

and therefore

$$\mathcal{L}_{\mathcal{D}}(f) \leq \mathcal{L}_S^{\theta}(f) + O\left(\frac{N\lg|H| + \lg(1/\delta)}{m} + \sqrt{\frac{N\lg|H| + \lg(1/\delta)}{m}\mathcal{L}_S^{\theta}(f)}\right)\,,$$

which concludes the proof of Theorem 1.