[Reviews · NeurIPS 2020]

Review 1

Summary and Contributions: This paper demonstrates that the k’th margin bound is inadequate in explaining the performance of gradient boosting. Accordingly, it provides a refined margin bound for gradient boosting.

Strengths: While I am not an expert on this topic, I feel the material is well motivated and developed. The empirical and theoretical results provide some new insights into gradient boosting.

Weaknesses: UPDATE: Thank you for your response, which addressed most of my concerns. The only issue is that only controlling the number of leaves can still be problematic since the depth of the also matters [1]. [1] Reyzin L, Schapire RE. How boosting the margin can also boost classifier complexity, ICML 2006. ============================= I have several concerns and questions: 1. Line 97: what does “same size” mean? We know that in order to make a fair comparison, we must make sure that the model complexity of base learners is the same. For decision trees, does it mean the same number of leaves? To me, the best way could be just using a decision stump as a base learner. 2. The empirical results are also not convincing to me: 1) the results are only averaged over three runs, which is insufficient to me. 2) I would also like to see the standard deviations of the average accuracies. 3) The experiments are only evaluated on one data set. To make the conclusion more convicting, the author should make a comparison on more data sets. 4) I was also wondering if the conclusion still holds with other base learners. 3. We know that AdaBoost is just a special case of gradient boosting with the exponential loss. Therefore, my feeling is that the analysis is just about the gradient boosting with different loss functions, not the behavior of gradient boosting itself.

Correctness: I feel the theoretical analysis is correct, though I didn’t go through the proofs of the theorems in the paper.

Clarity: The overall structure of this paper is clear to me.

Relation to Prior Work: The authors give a review and discussion on previous work. The theoretical analysis is novel to me.

Reproducibility: Yes

Additional Feedback:


Review 2

Summary and Contributions: The paper shows empirically that the margin of adaboost (AB) type ensembles grow with the number of combined classifiers (as known) but the margin of gradient boosting (GB) with combined base-classifiers diminishes (not known). However, the generalization accuracy of GB is slightly better than the generalization accuracy of AB. In addition, these observations contrast with the theoretical generalization bounds for these methods. Finally, the article derives a refined theoretical bound, more in line with the observed values.

Strengths: The paper is well written and sound (although I did not check proof sec 5). The observation made about GB is really interesting and deserves attention.

Weaknesses: UPDATE: I read the author's reply and I do not agree. In this text, I will focus on the two-class problem, {-1,+1}, for simplicity. First, GB combines regressors, and not classifiers, and their outputs cannot be normalized as classifiers. Second, the training of GB cannot be unlinked from the sigmoid as the pseudo-residuals are computed as the sigmoid times the class (Friedman 1999, section 4.5). In fact the output of the raw function of GB, that is F(x), tends to the log-odds ratio of the two classes. This quantity is unbounded even if the single regressors for the 2 class problem are bounded in [-1,+1]. Doing some simple experiments, one can check that the raw outputs of GB tend (in train) to +inf for +1 class instances and to -inf to -1 instances. This means that the pseudo-residuals tend to zero and the certainty of the ensemble tends to 100%. I also check that if one divides the raw outputs of GB by the number of base learners (or the sum of weights) the global outputs of the ensemble tend to 0 or to very low margins but I believe this normalization is not related to the training process of GB at all. The outputs of the base learners in GB tend to decrease with the size of the ensemble for the instances already "learned" with high confidence. This occurs precisely because GB learns an unnormalised additive expansion. I only have one concern about the article, that is, that it is not clear to me how the margin and output of GB is treated. GB uses regressors for each iteration. Hence, the outputs of the base classifiers of AD and GB are not directly comparable. It is true that the outputs of GB can be very small as shown in Fig.3 (I did not know, however, that most of them were so small). However, those outputs, after being linearly combined with the rest of the outputs, are passed through a sigmoid (for two class classification) that produces the final output of GB as a probability. In this context, it is unclear to me how the margin of GB is computed. It is computed as prob of correct class minus prob of the other class? This would produce the standard [-1, 1] margin range. In any case, the small values of GB may produce bigger changes after the sigmoid. Something that is different in AB where the -{-1, 1} outputs are directly summed up. From my point of view this should be clarified. I would also appreciate a deeper discussion about why the margin is reduced with the number of base rergessors in GB. Even if the outputs are small, that does not mean that the margins should reduce. GB is designed to reduce the residual of the outputs and reducing the pseudo-residuals to 0 and this would produce margin 1. I am willing to modify my score if this is clearly addressed.

Correctness: I believe everything is correct up to sec 4. I did not check the derivation of Sec5.

Clarity: Yes, the paper is well written

Relation to Prior Work: Prior work is properly discussed.

Reproducibility: Yes

Additional Feedback: The legend and lines of the plots are very thin and difficult to read. Missing reference in Line 69 A small detail: L2 says "great practical performance with little fine-tuning.". This is not really so, specially for GB. Only to navigate in the hundreds of hyper-params of LighGBM you can loss a great amount of time and in any case it needs to be tuned to get competitive results (learning rate, depth of tree, etc). I would just nuance the sentence. Another small detail: L19 says LightGBM is one of the best GB methods and I am not quite sure of that. I believe CatBoost and XGBoost or even standard GB are better than LightGBM in generalization capability. The benefit of LightGBM is its speed. The authors of [7] say it in the abstract of their article "LightGBM speeds up the training process of conventional GBDT by up to over 20 times while achieving *almost the same accuracy*."


Review 3

Summary and Contributions: The main result of this paper is a new generalization bound for voting classifiers (in the context of boosting). The bound is stated at Theorem 1 in Section 5 (a strange choice of the position of the main result). It improves margin-based bounds for boosting, that depends on theta^{-2}, where theta is the margin parameter. Here, the bound depends on the maximum between theta^{-1} and Bla*theta^{-2}, where Bla is some strange moment of the average value of |f(x)-h(x)|^2, where f is the voting classifier, x is a random example, and h is a random hypothesis sampled from the distribution induced by f. The authors show that the new bound correlate with empirical relative performance of AdaBoost vs. LightGBM on some benchmark datasets.

Strengths: Solid theoretical contribution that aims at explaining experimental results.

Weaknesses: This is a retrospect explanation of existing results, which always has the danger of over-fitting the theory to the results. A good theory should predict new phenomenon or yield new algorithms. The authors note this, but do not derive a new algorithm from the new bound. I'd give the paper a much higher score if the new bound would have yielded a new algorithm, which improves state-of-the-art in some aspect.

Correctness: Proofs seem correct.

Clarity: Yes.

Relation to Prior Work: Reasonably well.

Reproducibility: Yes

Additional Feedback:

[Author Response · NeurIPS 2020]

**Author Response ("Margins are Insufficient for Explaining Gradient Boosting")**

We thank the reviewers for the time and expertise invested in these reviews.

**Response to Reviewer 2.**    Addressing the reviewer's concerns, we wish to stress that while for Gradient Boosters (GB) the sigmoid function may be used to define the loss function and to transform the (raw) output prediction of a (learned) ensemble into a probability, the sigmoid function is not used when studying margin theory. We compute the margin of a an ensemble of base learners ($\{\alpha_i, h_i \mid h_i(x) \to [-1, 1]\}$) following classic margin theory (see Section 2). Loosely speaking, the margin of a point depends on the output of the voting classifier, and does not involve the sigmoid function. Formally, for data point $x_i$ the margin is $\frac{\sum_{j=1}^{n} \alpha_j h_j(x_i)}{\sum_{j=1}^{n} |\alpha_j|}$. Note the final remark of Section 2 of how we transform the output of GB algorithms, e.g. LightGBM, to fit this framework. This is unencumbered by the loss function minimized in training or how sigmoid may be used to transform the output of the ensemble into to a probability distribution by setting $P(y = 1 \mid x) = \sigma(\sum_{j=1}^{n} \alpha_j h_j(x_i))$. We also note that as opposed to the margins, this probability is not scale-invariant as scaling the weights or adding copies of the base learners provide more extreme probabilities.

Albeit unconnected to margin theory, the sigmoid function cannot make large changes to the output if the input is only changed by a small amount. In fact small changes to the input to the sigmoid function makes even smaller changes to the output since the derivative of the sigmoid function is at most 1/4 (and that is only at zero and the norm of derivative goes to zero as $|x|$ goes to infinity very fast).

Regarding the reason the margins are reduced with the number of base regressors in GB, in short this is due to the fact that gradient boosters may decide to focus on only a small subset of data in each iteration Consider the margin of a fixed data point $x$, and assume for simplicity that all weights ($\alpha_j$) are 1. Then the margin $x$ is $\frac{1}{n} \sum_i h_i(x)$, where $n$ is the number of base learners, which drops towards zero as $n$ increases unless $h_i(x)$ remains large (that is, close to 1) for all $i = 1, \ldots, n$. Hence, if each round only focuses on a small subset of data points, meaning that the new base learner only gives non-negligible prediction on a small subset, then the margin of the remaining points decreases. If this happens in each round of boosting, the result is a smaller and smaller margin distribution, and this is what happens in the experiments shown in our paper. We will update our description of this argument in Section 3 of our paper to make it more clear (alse see the arguments presented in Section 4 under Potentially Much Better).

We thank the reviewer for the specific comments as well. We will certainly take care of them for the final version.

**Response to Reviewer 1.**    As the reviewer suggests, the scope of our paper and theoretical contributions are more general than gradient boosted trees, as it applies to all voting classifiers irrespective of learning algorithm, including loss function optimized, and base learners. The paper investigates how the actual predictions of voting classifiers on the data may be very different than $\{-1, 1\}$, which is the standard assumption when proving margin bounds. Our theory, and margin theory in general, is orthogonal to specifics of the learning algorithm. We will think about ways of rephrasing our contributions to make it clearer that it is not only specific to gradient boosters with a concrete loss function. In practice, when conducting the numerical experiments, we chose to focus on comparing the classic AdaBoost algorithm where the base learner is restricted to map inputs to $\{-1, 1\}$ with Gradient Boosters that very often returns negligible predictions, as this exactly highlight and fits the phenomena we are investigating. We stress, however, that this is a special case, and the theoretical results we present are significantly broader.

For base learners, the same size means the same number of leaves (and no restriction on depth for both algorithms compared). We do not consider decision stumps as these are usually not used by GB in practice and lead to inferior performance (the default number of leaves for LightGBM is size 31, while XGBoost use a max depth default equal to 6). Furthermore, with decision stumps, the base classifiers are to weak for the phenomenon with small predictions to even occur. That is, with decision stumps, most predictions are very close to $\{-1, 1\}$. This is most likely due to the fact that decision stumps are incapable of "focusing" on a small subset of training points as discussed above. We will elaborate more on this in the final version and if space allows it, we will also include a histogram of predictions when using decision stumps.

In the supplemental material, submitted along with the paper, we included the same experiment on three more data sets to give 4 data sets of increasing size to analyze and demonstrate our new theoretical bound on. The observed behaviour was completely consistent. The mean validation error and standard deviation for the Forest Cover dataset example from the paper are (0.0298, 0.00037) for LightGBM and (0.0327, 0.00053) for AdaBoost. The standard deviation was so small that we chose to only show 3 runs on the plots. We will make sure to comment on this in a final version. For data sets included the in supp. material the mean, and standard deviation are as follows **Boone:** Lgb: mean=0.0574, std=0.00015, Ada: mean=0.0631, std=0.00068 **Higgs:** Lgb: mean=0.2530, std=0.00031, Ada: mean=0.2777, std=0.00009 **Diabetes:** Lgb: mean=0.2532, std=0.0255, Ada: mean=0.2692, std=0.0194 . for Boone and Higgs we used three runs and for Diabetes we used 10 (due to much smaller data size)

[Meta-Review · NeurIPS 2020]

R1 and R3 support acceptance underlying the interestingness of the results. R2 support rejects by mentioning that the results do not directly take into account some specificities of the gradient boosting (GB) learning algorithms in particular the problems of normalization of the regressors that have to be combined. That being said, the theory presented in the paper is fairly general, giving new insights on (gradient) boosting methods, it provides progress on margin bounds in both direction (lower+upper bounds) with respect to current state of the art. The wide use of (gradient) boosting methods make the paper interesting for the community. Based on these positive points, I recommend acceptance. However, the authors should consider revising their paper according to the following points: -The theory provided is rather general, not specific to GB, and must presented accordingly. Note that it is rather known/expected that existing bounds are not that tight, discussion could be improved on this point. -Consider to add a discussion that addresses the specificities of GB to answer the remarks raised by R2, and possibly the limits of the setup studied. -Consider to expand the discussion on the meaning of the proposed results: how do they make sense in practice and how to make use of them to develop new learning algorithms. -Consider to add the following references: *Lev Reyzin, Robert E. Schapire: How Boosting the Margin Can Also Boost Classifier Complexity. ICML 2006. Check in particular the experimental evaluation of the meaning of margin bounds *Liwei Wang, Masashi Sugiyama, Zhaoxiang Jing, Cheng Yang, Zhi-Hua Zhou, Jufu Feng: A Refined Margin Analysis for Boosting Algorithms via Equilibrium Margin. Journal of Machine Learning Research, vol 12, pages 1835-1863, 2011. *Robert E. Schapire, Yoram Singer: Improved Boosting Algorithms Using Confidence-rated Predictions. Machine Learning Journal, vol 37(3), pages 297-336, 1999 some issues mentioned in the paper were discussed already there *Leo Breiman. Prediction Games and Arcing Algorithms. Neural Computation, vol 11(7), pages 1493–1517, 1999. you should in particular make reference to the arc-gv algorithm (used in the experimental part of the Reyzin-Schapire-ICML'06 mentioned above)